# PIE-seq: identifying RNA-binding protein targets by dual RNA-deaminase editing and sequencing

Xiangbin Ruan [1], Kaining Hu [1] & Xiaochang Zhang [1] ✉

RNA-binding proteins (RBPs) are essential for gene regulation, but it remains a challenge to identify their RNA targets across cell types. Here we present PIE-Seq to investigate Protein-RNA Interaction with dual-deaminase Editing and Sequencing by conjugating C-to-U and A-to-I base editors to RBPs. We benchmark PIE-Seq and demonstrate its sensitivity in single cells, its application in the developing brain, and its scalability with 25 human RBPs. Bulk PIE-Seq identifies canonical binding features for RBPs such as PUM2 and NOVA1, and nominates additional target genes for most tested RBPs such as SRSF1 and TDP-43/TARDBP. Homologous RBPs frequently edit similar sequences and gene sets in PIE-Seq while different RBP families show distinct targets. Single-cell PIE-PUM2 uncovers comparable targets to bulk samples and applying PIE-PUM2 to the developing mouse neocortex identifies neural-progenitor- and neuron-specific target genes such as *App*. In summary, PIE-Seq provides an orthogonal approach and resource to uncover RBP targets in mice and human cells.

RNA-binding proteins (RBPs) and their interactions with RNAs are central to gene regulation[1]. RBPs such as FMR1 and TARDBP are causal for neurological disorders[2], and ablations of RBPs in mice have been shown to impair organ development such as the neocortex[3–7]. Conversely, mutations affecting regulatory pre-mRNA sequences have been shown to cause neurodevelopmental disorders[8–10]. Systematic identification of RBP target mRNAs in different cell and tissue types is key to understanding RBP functions and pathogenic mechanisms[11–14]. Methods based on UV crosslinking and immunoprecipitation (CLIP) have generated substantial insights[15–18], but their applications to cell types in intact tissues are hindered by the requirements of UV light penetration, high-quality antibodies and a large number of input cells.

RNA deaminases, such as Apobec1 and ADAR2, bind to RNA targets with auxiliary proteins and introduce C-to-U and A-to-I nucleobase changes, respectively[19,20]. Fusing RNA deaminase domains with RBPs of interest would guide RNA editors to target RNAs and introduce nucleotide changes that can be detected by RNA sequencing (RNA-Seq). In 2016, the catalytic domain of *Drosophila* ADAR was fused with RBPs to identify mRNA targets (TRIBE) through A-to-I editing in fly cells[21]. An E488Q mutation in the ADAR C-terminal domain has been introduced to enhance the editing efficiency in HyperTRIBE, which uncovered 4E-BP

and MUSASHI-2 targets in human cells[22,23]. Conceptually similar to TRIBE, the C-to-U deaminase Apobec1 was conjugated to a YTH domain to detect N6-Methyladenosine (m6A) sites through RNA-Seq[24], and Apobec1-RBP-fusion proteins (STAMP) have been utilized to identify target mRNAs of RBFOX2, TIA1, SLBP and ribosome-subunits in cultured cells[25]. These studies indicate that RNA editing is a promising approach to studying protein–RNA interaction. While the single deaminase approaches bypass some of the limits for CLIP-based methods, their intrinsic bias toward A- or C-nucleobases, relative editing efficiency, scalability, and in vivo applications remain to be addressed[26].

Here we present PIE-Seq utilizing dual deaminases to enhance target detection: we fuse RBPs to A-to-I and C-to-U RNA deaminase domains and directly uncover RBP target genes by RNA-Seq. Dual deaminases in PIE-PUM2 enhance target discovery when compared with single deaminases. PIE-Seq successfully identifies target genes and binding motifs for PUM2, NOVA1, and 23 additional RBPs, demonstrating its consistency and scalability. We further show the applications of PIE-Seq to single cells and cell types isolated from the developing mouse neocortex. This study presents PIE-Seq as an orthogonal approach to studying protein–RNA interaction and provides a resource to investigate the target genes of 25 RBPs.

[1]Department of Human Genetics and The Neuroscience Institute, University of Chicago, Chicago, IL, USA. ✉e-mail: xczhang@uchicago.edu

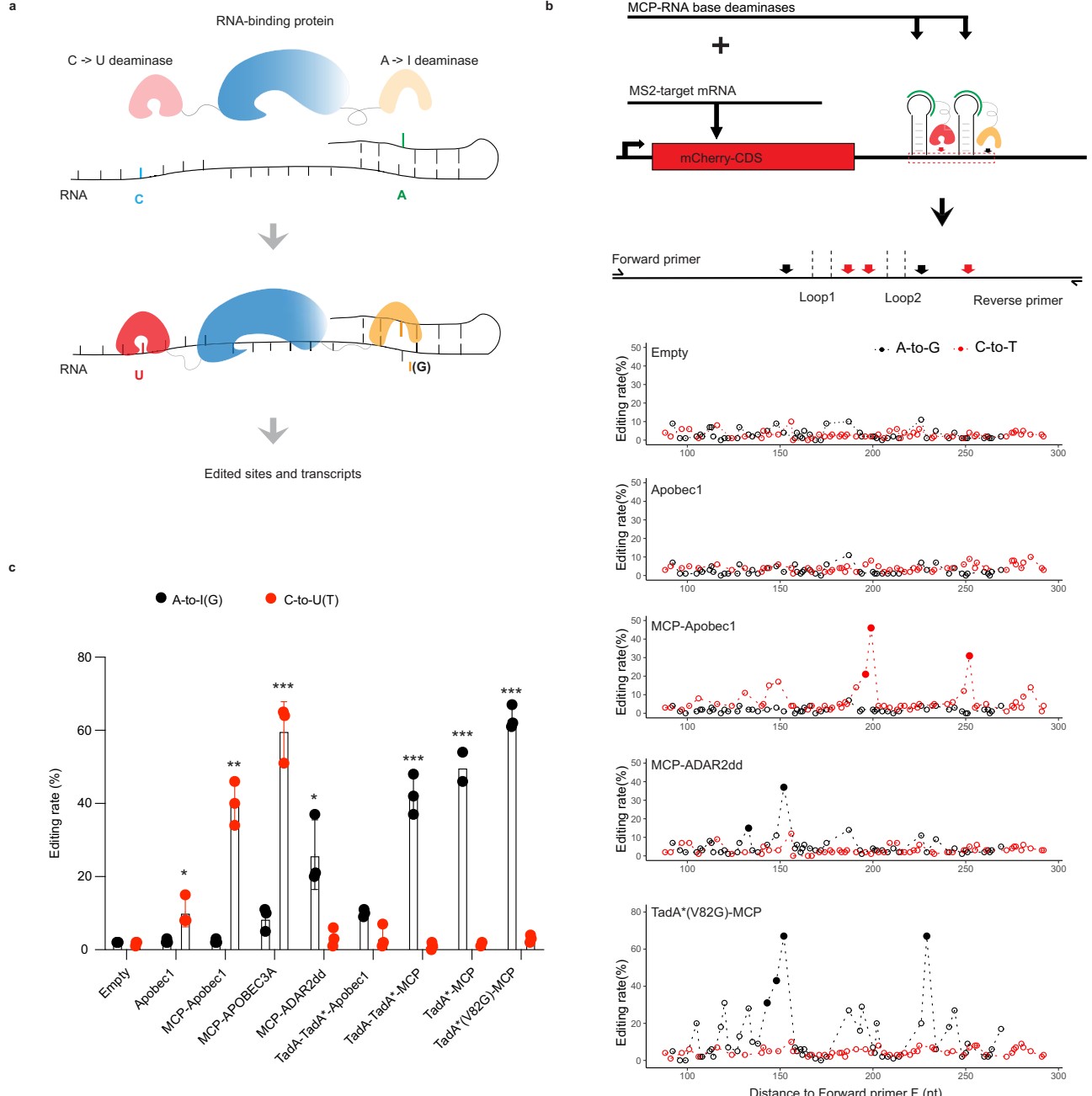

**Fig. 1 | The PIE-Seq design and evaluation of deaminase modules. a** Rationale of PIE-seq: when fused with RBPs, RNA deaminase domains introduce C-to-U and A-to-I editing sites in proximity sequences to RBP binding sites. **b** Evaluation of deaminase activities with MCP-MS2. Apobec1, ADAR2dd, or TadA* was fused with MCP and co-expressed with mCherry-2xMS2 in HEK293FT cells. Sanger sequencing results show C-to-U(T) and A-to-I(G) RNA mutations surrounding MS2 sites. Red dots, C-to-T; black dots, A-to-G; solid dots were significant and hollow dots were insignificant. Quantified with the MultiEditR package[81]. **c** Quantification of deaminase editing rates showing the editing site at the highest edited position in each group (Fig. 1b and Supplementary Fig. 1a). Each group with $n = 3$ biologically replicates except $n = 2$ for TadA*-MCP groups. The student's $t$-test with two-tailed distribution was performed: For C-to-U (T) sites, Apobec1 to Empty ($p = 0.021$), MCP-Apobec1 to Apobec1($p = 0.002$), and MCP-APOBEC3A to Apobec1 ($p < 0.001$); For A-to-I (G) sites, MCP-ADAR2dd to Empty ($p = 0.012$), other TadA*-MCP constructs to TadA-TadA*-Apobec1 ($p < 0.001$). * indicates $p < 0.05$, ** indicates $p < 0.01$, and *** indicates $p < 0.001$. Data were presented as mean values ± SD. Source data are provided as a Source Data file.

## Results

### The rationale of PIE-Seq and identification of RNA deaminase modules

C-to-U and A-to-I editing can potentially complement each other in different sequence or structure contexts and enhance target discovery. PIE-RBPs were constructed by fusing C-to-U and A-to-I RNA deaminase domains to RBPs with flexible linkers. Upon RBP binding to its RNA targets, the dual deaminases in PIE-RBPs are expected to

catalyze C-to-U and A-to-I editing in adjacent sequences, which can be readily detected by bulk or single-cell RNA-Seq (Fig. 1a).

We utilized the protein–RNA-binding pair MCP-MS2[27] to determine RNA deaminase activities for rat Apobec1, human APOBEC3A, human ADAR2 deaminase domain (ADAR2dd)[28], and codon-optimized TadA variants (Methods). Apobec proteins introduce C-to-U editing in an AU-rich context[29]; the ADAR2 deaminase domain prefers to edit bulged A sites in double-stranded RNAs (dsRNAs)[30] and evolved TadA

variants (TadA* and TadA*(V82G)) have been reported to retain transcriptome-wide RNA editing activity[31]. We fused MCP to the deaminases and inserted a 2xMS2 sequence downstream of the mCherry coding sequence (CDS) to make the mCherry-2xMS2 reporter (Methods). When MCP-Apobec1 was co-expressed with mCherry-2xMS2, we detected significantly higher C-to-U editing rates (18–46%) surrounding the MS2 sites than the empty control (1.7–5.3%) or the non-targeting Apobec1 control (7.3–14.3%, Fig. 1b, c). MCP-APOBEC3A, but not the K30R/Y132G[32], introduced two C-to-U editing events flanking the 2xMS2 sequences (Supplementary Fig. 1a, over 50% editing rate). The ADAR2dd retained A-to-I editing activity when it was fused to MCP (Fig. 1b, c). TadA-TadA*-MCP, TadA*-MCP or TadA*(V82G)-MCP showed higher editing rates than ADAR2dd with excess editing sites throughout the 2xMS2 region (Fig. 1b, c and Supplementary Fig. 1a). On the reporter construct, we observed that the A-to-I (ADARD2dd) editing site was at 26nt, and the C-to-U (Apobec1) editing site was at 73nt; both sites were in proximity to the two well-defined RNA stem-loops (27-67nt and 77-95nt, Supplementary Fig. 1b). To exclude the possibility of DNA editing, we directly amplified the 2xMS2 sequence in DNA extracted from transfected cells and found no significant mutation (Supplementary Fig. 1c), suggesting that the base editing is RNA-specific. These results indicate that Apobec1, APOBEC3A, ADAR2dd, and TadA* retained RNA deaminase activities when fused with MCP and introduced reproducible base changes nearby MS2 sites, making them candidate enzymatic modules for PIE-Seq.

## Benchmarking PIE-Seq with PUM2

The PIE-Seq workflow starts with cloning and expression of PIE-RBPs, followed by RNA Sequencing, variants calling, and statistical analyses of edited nucleobases and genes (Fig. 2a). We first benchmarked PIE-Seq with PUM2, which plays essential roles in body size control and neural development[33,34]. Mechanistically, PUM2 binds to the UGUA-NAUA sequence (or Pumilio Response Element, PRE) in 3′ untranslated regions (3′ UTRs) through its Pumilio homology domain and regulates gene expression[35,36]. We fused Apobec1 and ADAR2dd to the N- and C-terminus of PUM2, with XTEN[37] and Glycine-Serine (GS) peptide-linkers to make the fusion protein conformationally flexible (Figs. 1a, 2, Methods). We used two controls for PIE-PUM2: an empty-vector control without deaminases (empty control) to filter out background genetic variations and editing sites; and an Apobec1-ADAR2dd deaminase-only control (APAD or non-targeting control) to filter out stochastic editing events. When expressed in HEK293FT cells, the PIE-PUM2 fusion protein was enriched in the cytoplasm and displayed no observable adverse effect on cell growth or survival (Fig. 2b).

We harvested cells 48 h after transfection and performed stranded RNA-Seq for polyadenylated mRNAs. Through RNA variants calling with the JACUSA2[38], we quantified the number of editing sites per gene in the APAD non-targeting control and PIE-PUM2 after filtering out background variations. For A-to-I editing, more than 70% of edited genes had one or two sites in both APAD and PIE-PUM2 (Supplementary Fig. 2a). For C-to-U editing, 85% of edited genes in APAD and 31% in PIE-PUM2 showed potential hyper-editing events[39] (defined here as 20 or more editing sites per gene), suggesting that stochastic C-to-U editing events in APAD were reduced by PIE-PUM2 (Fig. 2c, d).

To identify acute editing sites, we expressed PIE-PUM2 for 24 h and sorted EGFP-positive cells for further analysis. Taking the bona fide PUM2 target gene CDKN1B as an example[40], PIE-PUM2 showed a 55% editing rate in EGFP high cells and non-significant editing in EGFP low cells (Fig. 2e and Supplementary Fig. 2b). In the 24-hour EGFP high-positive cells, the number of potential hyper-editing events was significantly lowered than in the 48-hour samples (Fig. 2c, d). The ratio of reproducible editing sites (editing rate >5%) between PIE-PUM2 replicates was significantly higher than that of the APAD group (Supplementary Fig. 2c), and PIE-PUM2 edited different sets of nucleobases from the APAD control (Supplementary Fig. 2d, e). These results

indicate that transient PIE-PUM2 is sufficient to introduce A-to-I and C-to-U editing in target transcripts.

Using PIE-PUM2 as an example, we defined PIE-Seq target sites and target genes with the following criteria (see details in Methods): (1) We designated editing sites as target sites if they show significantly higher mutation frequencies in PIE-RBP than the APAD group (by editing rate fold change and z-score generated by JACUSA2, Methods); (2) The genes with target sites (C-to-U or A-to-I) were considered as target genes. In total, PIE-PUM2 uncovered 1975 target genes with 1811 genes reproduced between replicates (Fig. 2f). Among them, 250 genes had both C-to-U and A-to-I target sites, 1156 genes had C-to-U only, and 569 genes were uniquely marked by A-to-I (Fig. 2g). The PIE-PUM2 target genes (Fig. 2g, k) significantly intersected with the genes identified by PUM2 PAR-CLIP[15] and PUM2 HITS-CLIP[41] in HEK293T cells, and PUM2 RNA Immunoprecipitation (RIP)-Seq in Hela cells[42]. These results indicate that the dual deaminases uncovered PUM2 target genes and reduced false negatives by single-base editors.

In parallel, we cloned PIE^TadA*-PUM2 by replacing ADAR2dd CDS in PIE-PUM2 with TadA*(V82G) (Fig. 2h), which showed high editing rates in the MCP-MS2 assay (Fig. 1c). PIE^TadA*-PUM2 identified 375 target genes, over half of which (205 genes) were shared with PAR-CLIP PUM2 targets (Fig. 2h). The limited number of PIE^TadA*-PUM2 target genes suggest either incompatible protein conformations or potential deleterious effects of excess editing sites introduced by TadA*(V82G, Fig. 1b). We decided to use Apobec1 and ADAR2dd in the following PIE-Seq experiments.

To compare the PIE-Seq dual-deaminase approach with single-deaminase methods such as TRIBE and STAMP, we cloned Apobec1-PUM2 (AP-PUM2) and PUM2-ADAR2dd (PUM2-AD), with AP-mCherry and mCherry-AD as deaminase-controls, respectively (Fig. 2i). The proportion of AP-PUM2 target genes shared with PUM2 PAR-CLIP is comparable to that in PIE-PUM2 (46.6% in AP-PUM2 versus 49.5% in PIE-PUM2 C-to-U), and a smaller proportion of PUM2-AD target genes than that of PIE-PUM2 were uncovered in PUM2 PAR-CLIP (34.3% in PUM2-AD versus 41.8% in PIE-PUM2 A-to-I, Fig. 2g, i). Integrated analysis showed that significant numbers of A-to-I and C-to-U target sites and target genes were shared between PIE-PUM2 and PUM2-AD (565 sites and 594 genes) or between PIE-PUM2 and AP-PUM2 groups (449 sites and 634 genes) (Fig. 2j and Supplementary Fig. 2f). The majority (1707/1975) of PIE-PUM2 target genes were cross-validated by AP-PUM2, PUM2-AD, PUM2 RIP-Seq, or different CLIP data (Fig. 2k). These results suggest that dual deaminases in PIE-RBP maintained the deaminase activity of Apobec1 and ADAR2dd, and jointly discovered target genes.

## PIE-Seq identifies PUM2 binding motifs

The nature of C-to-U and A-to-I base editing brings in confounding bias for Cs and As at the editing sites, which would profoundly skew the sequence motif analysis. In addition, Apobec1 preferably edited the C in ACH sequences, and ADAR2dd edited the A in WAS sequences in the non-targeting Apobec1-ADAR2dd controls (Supplementary Fig. 3a). To overcome these intrinsic limits, we excluded the edited C or A base and explored the enriched RNA motifs either 50-nt upstream or downstream of the edited sites separately.

We then asked whether PIE-PUM2 target sites were associated with the canonical PUM2 binding motif UGUANAUA. RNA sequences downstream of PIE-PUM2 target sites (588/2050 C-to-U and 94/918 A-to-I target sites) showed enriched UGUAM and UGUAHA motifs which represent the core PRE sequence (Fig. 3a). Interestingly, 237 target sites were within 5-nt upstream of the UGUANA motif, indicating that PIE-PUM2 fusion protein prefers to edit nearby bases upstream of the binding site (Fig. 3b). This number is probably underestimated considering that the nucleobases at the editing sites were excluded from motif analyses. Noticeably, there were significantly fewer editing sites within 10-nt downstream of the first U of the UGUANA motif (0 editing

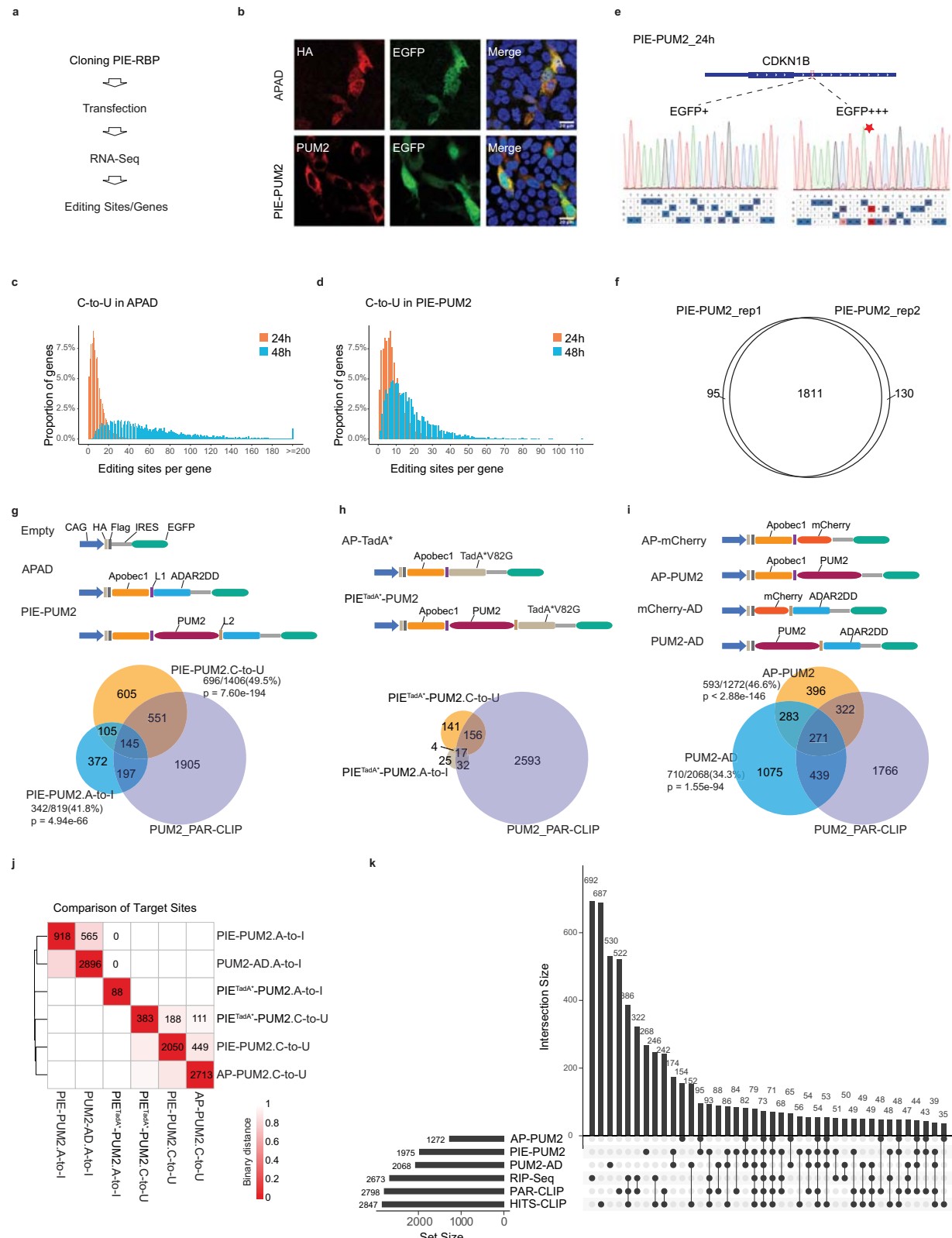

sites within the first 5 nucleotides), suggesting that PUM2 binding to RNA precluded ten nucleotides from deaminases (Fig. 3b). There were 698 out of 1975 (>35%) PUM2 target genes showing UGUANA motif within ±50-nt of target sites; target genes with both A-to-I and C-to-U editing sites showed higher enrichment of PRE motifs (53%) than A-to-I (16%) or C-to-U (41%) only targets. Furthermore, the proportion of these 698 genes overlapping with PUM2 PAR-CLIP target genes was

slightly higher than that of total 1975 PIE-PUM2 target genes (53 vs 45%, Supplementary Fig. 3b). PIE-PUM2 target sites were enriched in the 3′ UTR with 15% sites in CDS (Supplementary Fig. 3c), which is consistent with a previous report[43]. As an example, PIE-PUM2 identified significant target sites in *TOP2A*, encoding a DNA topoisomerase critical for DNA replication[44,45], and such target sites were surrounded by PRE and multiple CLIP-Seq peaks (Fig. 3c). These results indicate that PIE-PUM2

**Fig. 2 | Deaminase modules are active in PIE-Seq fusion proteins. a** The PIE-Seq workflow. **b** Immunostaining of HA tag, PUM2, and EGFP in HEK293FT cells that were fixed 48 h after APAD or PIE-PUM2 transfection. Hoechst staining for nuclei with blue signal in merged images. Scale bar, 20 μm. Representative images from two independent experiments. **c**, **d** Bar plots showing proportions of genes with different numbers of C-to-U editing sites in APAD (**c**) and PIE-PUM2 (**d**) 24 or 48 h after transfection. **e** Sanger sequencing results showing PIE-PUM2 editing rates on *CDKN1B* 3′ UTR in EGFP high (EGFP + ++, top 40%) or low (EGFP+, bottom 40%) HEK293FT cells 24 h after transfection. **f** Venn diagrams showing that PUM2 target genes were highly reproducible between two PIE-PUM2 biological replicates. **g**–**i** Upper panel: Schematics of PIE-PUM2 and related control constructs (**g**),

PIE^TadA*-PUM2 and control constructs (**h**), and single deaminase AP-PUM2, PUM2-AD and corresponding control constructs (**i**). Lower panel: Venn diagrams showing C-to-U and A-to-I target genes identified in the corresponding constructs. L1, linker 1 for XTEN linker; L2, linker 2 for (GGS)$_{X3}$GG linker. A hypergeometric distribution test was applied for comparison. **j** Heatmaps showing the Euclidean distance among target sites identified by PIE-PUM2, PIE^TadA*-PUM2, AP-PUM2, and PUM2-AD groups. A smaller binary distance value indicates a closer distance. The numbers indicate how many target genes were shared between sample pairs. **k** An UpSetR plot comparing target genes between PIE-PUM2, AP-PUM2, PUM2-AD and PUM2 PAR-CLIP, RIP-Seq and HITS-CLIP.

identified PUM2 binding motifs and target genes, and the edited RNA bases tended to be close to the PUM2 binding site.

## Inducible PIE-Seq

We further introduced Tet-On Doxycycline (Dox)-inducible expression system[46] to APAD and PIE-PUM2 to modulate the expression levels. The RNA-Seq and Western blot analyses confirmed that the levels of induced PUM2 expression were significantly lower than those observed in the transient expression group (Supplementary Fig. 3d, e). The target genes identified by inducible PIE-PUM2 largely overlapped with transient targets, though significantly fewer genes were identified by the induced groups (Fig. 3d). Importantly, the shared target sites between PIE-PUM2 and induced groups showed significantly higher confidence scores during target site calling (*p* value <2.2e −16 for both low and high Dox groups) compared to those not shared in PIE-PUM2 (Fig. 3e). The PRE motif was enriched surrounding target sites in the Dox high induction PIE-PUM2 group (Supplementary Fig. 3f). These results suggest that PIE-Seq can be integrated with an inducible expression system to uncover RBP targets.

We performed RNA Immunoprecipitation (RIP) and qPCR to validate PIE-PUM2 targets. We selected 42 high-confidence target genes (see Methods), including 22 shown in PAR-CLIP data, for independent validation through RIP-qPCR. Twenty-seven genes showed significant enrichment in the PUM2 RIP group when compared to the IgG control, with 17 shown in PAR-CLIP, three unique to transient PIE-PUM2, and seven shared only between transient PIE-PUM2 and Tet-on sets (Fig. 3f). These results indicate that PIE-Seq identified known and previously unknown PUM2 target genes in human cells.

## PIE-Seq in single cells

To explore the possibility of adapting PIE-Seq to single-cell analysis, we first examined the MCP-Apobec1 with the MCP-MS2 system. We co-expressed MCP-Apobec1 with the mCherry-2xMS2 reporter and sorted mCherry/EGFP double-positive cells. We amplified transcripts in sorted single or ten cells with the SMART-Seq2 protocol[47], and consistently detected RNA editing sites in 10-cell and single-cell MCP-Apobec1 samples but not in the Apobec1 non-targeting controls (Supplementary Fig. 4a). Interestingly, the 10-cell and single-cell samples showed higher C-to-U editing ratios than bulk samples. These results indicate that MCP-Apobec1 introduced C-to-U editing to sequences close to the 2xMS2 binding site in single cells.

We then expressed PIE-PUM2 or the APAD control in HEK293FT cells, isolated cells with flow cytometry, and prepared RNA sequencing libraries with SMART-Seq2 (Fig. 4a). We observed consistent C-to-U target sites in the *CDKN1B* gene in sorted PIE-PUM2 positive single-cell and ten-cell samples but not in the APAD non-targeting control (Fig. 4b), which was further validated by Sanger sequencing (Supplementary Fig. 4b). In both single- and ten-cell samples, we found that PIE-Seq detected larger numbers of PUM2 target genes than those identified in bulk PIE-PUM2 (Supplementary Fig. 4c), which was probably caused by high PIE-PUM2 expression and higher read depth than bulk samples (80–110 million reads versus 30 million reads per replicate). We down-sampled the single-cell and 10-cell data

and found that even with a 75% reduction in read depth (25% down-sampling to make the coverage comparable to bulk samples), we still identified over 67 and 63% of the target genes in single-cell and 10-cell samples, respectively (Fig. 4c and Supplementary Fig. 4c, d). Single-cell and 10-cell PIE-PUM2 target genes were consistent with bulk PIE-PUM2 and PUM2 PAR-CLIP targets (Fig. 4c and Supplementary Fig. 4c, e). These results indicate that PIE-Seq is applicable to identify RBP target mRNAs in single cells.

## Identification of cell-type-specific PUM2 targets in the developing mouse neocortex

RBPs play essential functions in brain development through spatial and temporal regulation of diverse RNA species[4,5,14]. To address the challenge of identifying RBP target genes among brain cell types, we delivered PIE-PUM2 constructs carrying an EGFP cassette into embryonic day 13.5 (E13.5) mouse dorsal forebrains using in utero electroporation (IUE). We dissected the electroporated dorsal cortical tissues at E15.5 and isolated EGFP-positive neural-progenitor cells (NPCs, CD133+) and neurons (CD24+)[48,49] for RNA Sequencing (Fig. 5a, b and Supplementary Fig. 5a). Differential gene expression analysis confirmed the expression of cell identity genes for neurons and NPCs in sorted samples (Supplementary Fig. 5b).

PIE-PUM2 identified 448 and 598 target genes in mouse cortical neurons and NPCs, respectively, with 72 genes shared between them (Fig. 5c). Gene Ontology analysis of neuronal PIE-PUM2 target genes showed enriched terms such as positive regulation of transcription, while PIE-PUM2 NPC target genes were enriched in cell migration and chromatin organization (Supplementary Fig. 5c). Interestingly, PIE-PUM2 uncovered a C-to-U target site on the 3′ UTR of *App* (amyloid beta precursor) nearby a PRE motif in the NPC group but not in the neuron group (Fig. 5d). Although the levels of *App* mRNA expression were comparable between NPCs and neurons (Fig. 5e), we observed high Pum2 protein levels in NPCs in the ventricular zone and higher App protein levels in the intermediate zone (Fig. 5f). Reanalysis of a previous Pum2 dataset uncovered iCLIP peaks at the *App* 3′ UTR in wild type but not *Pum2* knockout mouse brains[34] (Fig. 5d). These observations suggest that Pum2 plays a role in inhibiting App translation and indicate that PIE-Seq can detect cell-type-specific PUM2 target genes in the developing mouse brain.

## PIE-Seq reveals binding motifs and target genes of SR proteins

After benchmarking PIE-Seq with PUM2, we seek to explore the scalability of the method with additional RBPs. SR proteins play important roles in pre-mRNA splicing with arginine (R) and serine (S) enriched RS domains and variable RNA recognition motifs (RRM)[50,51]. We applied PIE-Seq to SRSF1/2/3 proteins and sought to determine whether PIE-Seq can distinguish RNA-binding features of homologous RBPs. SRSF1, SRSF2 and SRSF3 have the RRM domains at the N-terminus, RS domains at the C-terminus, and variable sequences in the middle (Fig. 6a). To minimize the impact of stochastic editing while applying PIE-Seq to SRSF1/2/3 and additional RBPs below, we included two more controls in addition to AP-AD: mCherry and the deactivated PspCas13b were inserted between Apobec1 and ADAR2dd to create the AP-

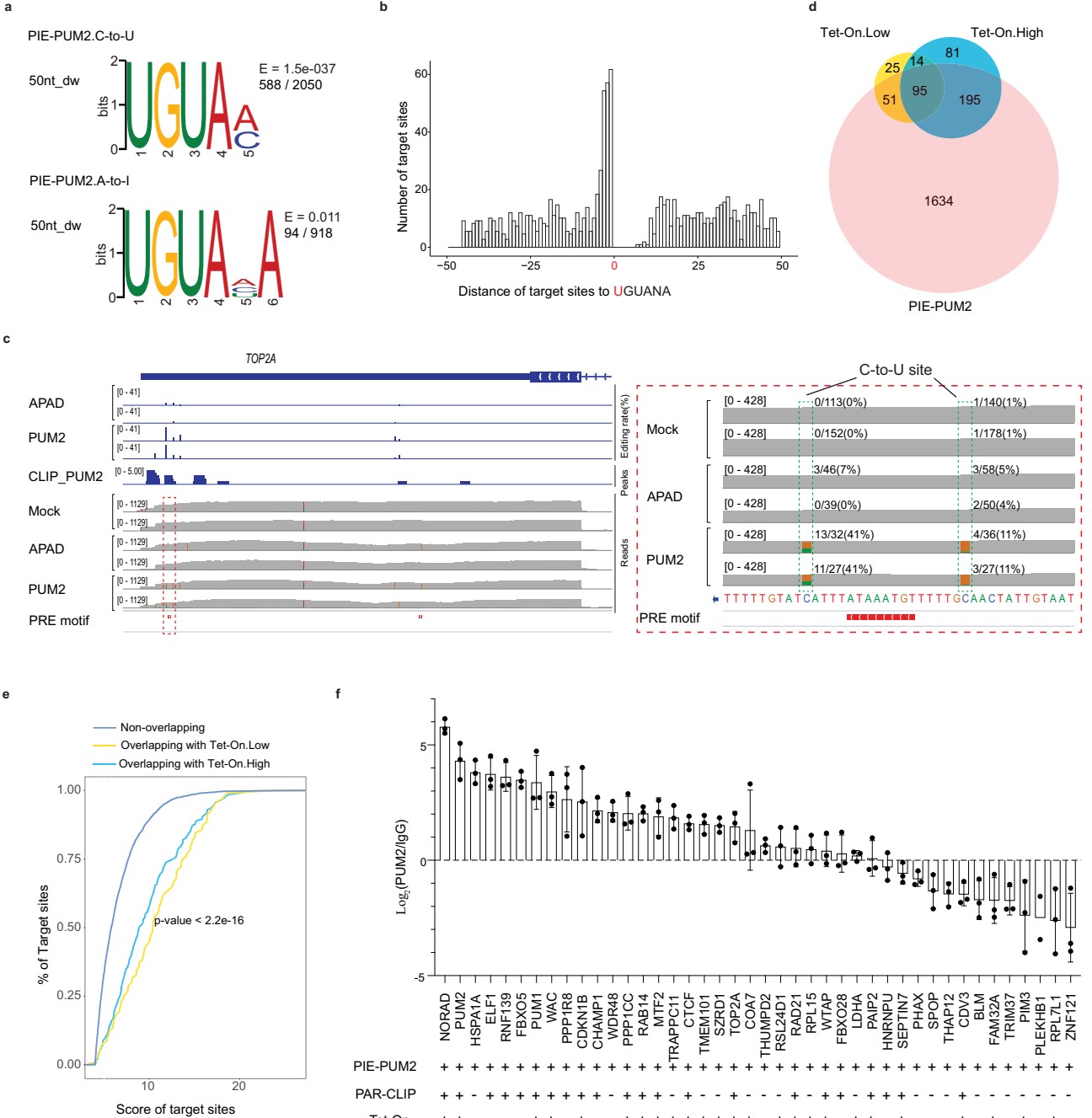

**Fig. 3 | PIE-Seq identifies the PUM2 binding motif and target genes. a** The most enriched motifs within the ±50-nt region of C-to-U and A-to-I PIE-PUM2 target sites. The E-value is the number of candidate motifs times the enrichment *p* value, which is tested by Fisher's exact test. **b** Distribution of PIE-PUM2 target sites flanking the UGUANA motif (the first Uracil was designated as 0). **c** Integrative genomics viewer (IGV) and bedGraph tracks showing RNA-Seq reads, PIE-Seq editing sites, PAR-CLIP peaks, and PRE sequences in *TOP2A* 3′ UTR. The dashed rectangle was zoomed in on the right panel. **d** The target genes identified by inducible PIE-PUM2 were largely confirmed in transient PIE-PUM2. **e** Empirical cumulative distribution function (ECDF) plot showing target sites shared between transient and inducible PIE-PUM2 groups had higher *z*-scores than non-overlapping targets in transient PIE-PUM2. The *p* value is calculated by the ECDF function in R. **f** PUM2 RIP-qPCR results in HEK293FT cells. Each gene was quantified with *n* = 3 (except *n* = 2 for *PLEKHB1*) independent biological samples. ± represent whether the gene exists in the indicated group or not. "Tet-On" is the union of Tet-On.Low and Tet-On.High groups. Data were presented as mean values ± SD. Source data are provided as a Source Data file.

mCherry-AD and AP-dCas13b-AD non-targeting controls, respectively. We combined the variant calling data from independent AP-AD, AP-mCherry-AD and AP-dCas13b-AD controls as the combined negative control (APADcom) group for the analyses below (Supplementary Fig. 6a).

Using PIE-SRSF1/2/3 constructs (Supplementary Fig. 6a), we identified 4104, 4415 and 2784 target genes in HEK293FT cells for SRSF1, SRSF2 and SRSF3, respectively. 35% of PIE-SRSF1/2/3 target genes (1955/5571) were shared among the three, and over 70% of SRSF1 and SRSF2 targets were shared (3293/4104 and 3293/4415, Fig. 6b). Noticeably, over 90% of these PIE-SRSF1/2/3 target genes (APADcom control) were reproducible when APAD was used as the control (Supplementary Fig. 6b), suggesting it is sufficient to use either the single APAD or the combined APADcom control in practice. The target sites

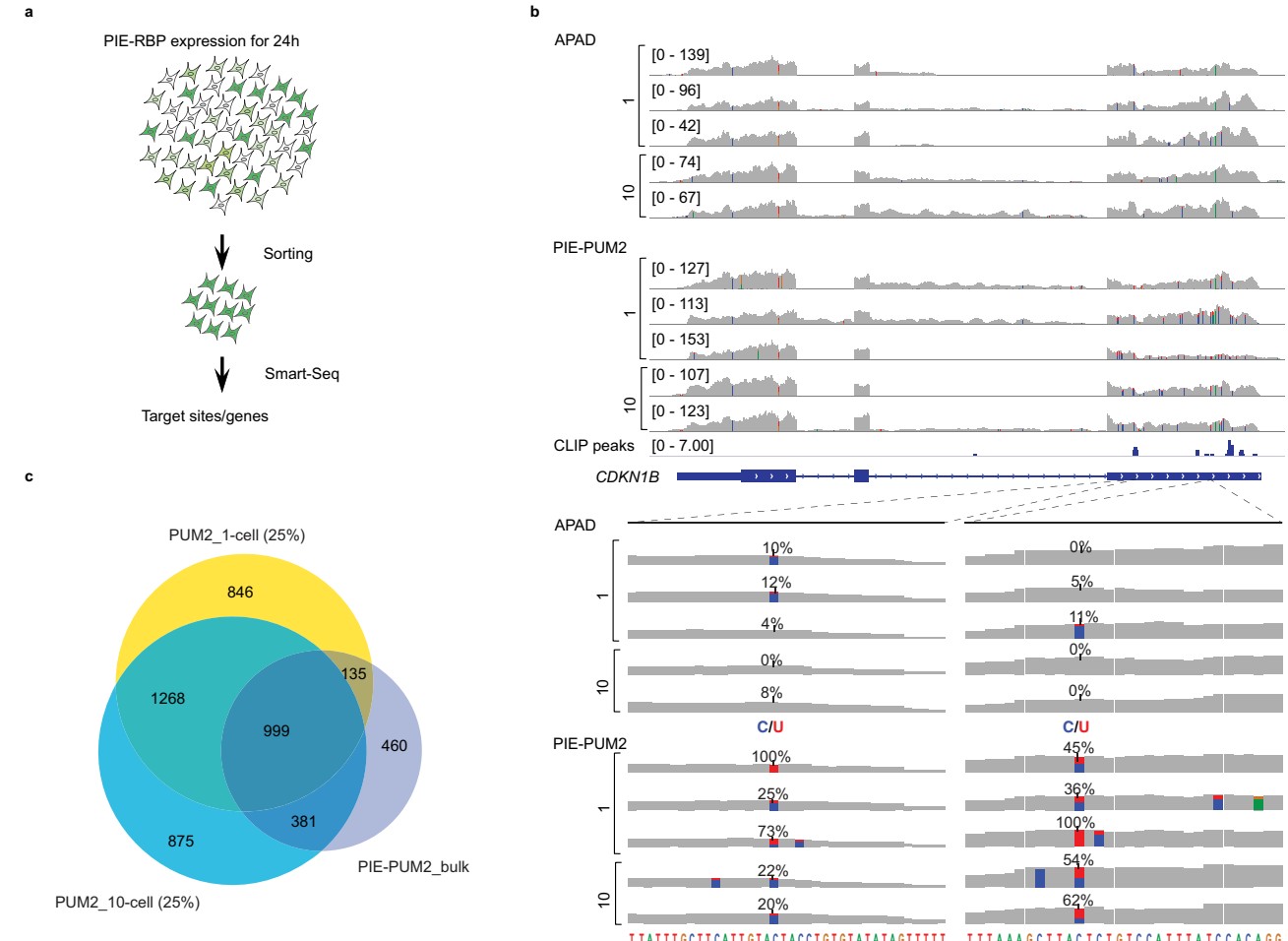

**Fig. 4 | PIE-PUM2 uncovers mRNA targets in single cells. a** The workflow of PIE-Seq for 10-cell and single-cell samples. **b** IGV browser tracks of the *CDKN1B* gene showing PIE-PUM2 editing sites in single-cell and ten-cell samples. C-to-U editing rates over 10% are indicated as blue (Cytosine) and red (Uracil). **c** Venn diagrams showing PUM2 targets identified in single-cell and ten-cell samples (25% down-sampling, details in supplementary Fig. 4c–e) largely overlapped with those of bulk PIE-PUM2.

in PIE-SRSF1/2/3 were enriched in coding regions when compared with the APADcom control (Supplementary Fig. 6c). Motif analyses showed that SRSF1 and SRSF2 shared the GAWGV motif upstream of target sites, but SRSF3 prefers to bind to distinct CHNC or CDNC motifs (Fig. 6c). There were two target site peaks surrounding the GAWGV motif for both SRSF1 and SRSF2, one within 2 ~ 10-nt upstream and the other at 1-nt downstream (Fig. 6d). These results suggest that PIE-Seq identified shared and distinct targets for homologous SRSF1/2/3 proteins.

**PIE-Seq uncovers binding features and target genes for 25 RBPs**
To test the scalability of PIE-Seq and gain more insights into RBP functions, we applied PIE-Seq to 21 additional human RBPs, including NOVA1, CELF1/2/4, FMR1, FUBP1, IGF2BP1/2, KHDRBS1/2/3, LIN28A/B, QKI isoforms, STAU2, TARDBP, YTHDC1/2 and YTHDF1/2, many of which are associated with neural development or neurological disorders[52–54]. In total, we performed PIE-Seq for 25 RBPs in HEK293FT cells (Fig. 7) and called target sites and target genes (Supplementary Data 2). We performed the principal component analysis (PCA) using editing rates for all raw editing sites (Supplementary Fig. 7a), all target sites (Fig. 7a), or all target genes (Fig. 7b). In all three analyses, RBPs showed a close association with their homologous proteins except for SRSF3 (Fig. 7b), likely due to its different binding preference from SRSF1/2 (Fig. 6). These results suggest that: (1) different RBP families have largely different sets of target sites and target

genes, while homologous RBPs tend to share common targets; (2) PIE-Seq targets are driven mainly by the expression of RBPs instead of deaminase modules.

We seek to determine whether PIE-Seq uncovers RBP binding motifs by looking for enriched sequences in the 50-nt flanking regions of target sites. Overall, the most enriched motifs identified by PIE-Seq showed three prominent features: (1) homologous RBPs showed highly similar enriched motifs (Fig. 7c); (2) Enriched sequence motifs were largely consistent with previously reported binding motifs if there were any (Supplementary Data 3); (3) Enriched sequence motifs were concentrated around the target sites (Fig. 7d and Supplementary Fig. 7b). These results indicate that PIE-Seq is robust in identifying RBP binding sites.

We use NOVA1 and YTH proteins here as examples to further illustrate the power and limitations of the PIE-Seq approach and discuss additional RBPs below (Supplementary Fig. 8 and Supplementary Note 1). Nova1 is essential for mouse survival[55] and extensive CLIP-Seq studies in the brain have shown that Nova1 binds YCAY sequences to regulate RNA splicing[16,56,57]. We studied the human NOVA1 and identified the bona fide YCAY as the most enriched motif surrounding both C-to-U and A-to-I target sites (Fig. 7c). The target sites displayed bimodal distribution surrounding the YCAY motif (−10 to −3nt and +6 to +13nt, Fig. 7d and Supplementary Fig. 7b). 1190 of 1965 target genes identified by PIE-NOVA1 were confirmed by both NOVA1 HITS-CLIP of E18.5 mouse brains and eCLIP-Seq of human cerebral organoids

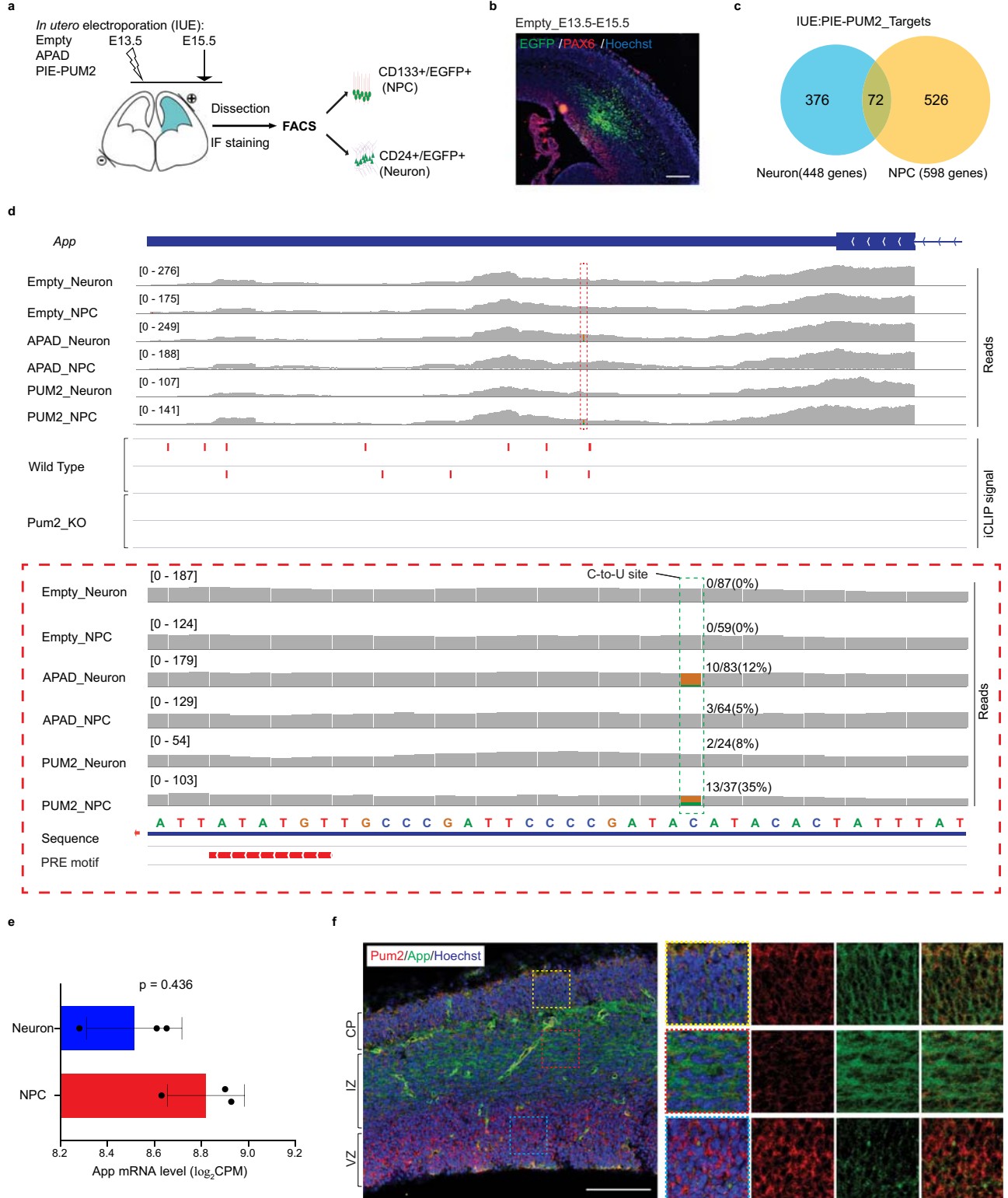

**Fig. 5 | Applying PIE-PUM2 to cortical progenitors and neurons identifies App as one of its targets. a** The IUE and PIE-Seq workflow in the developing mouse neocortex. **b** Immunostaining results showing the distribution of EGFP-positive cells in the E15.5 dorsal cortex. The germinal zone is indicated by anti-PAX6 (red). Hoechst (blue) labels DNA. Scale bar, 200 μm. Representative image from three independent experiments. **c** Venn diagrams showing PUM2 target genes in neurons and NPCs. **d** RNA-Seq and iCLIP tracks of the *App* gene showing the NPC-specific PIE-PUM2 target site and Pum2 iCLIP signals from newborn mouse brains, respectively. **e** Differential gene expression analysis showing comparable *App* mRNA expression levels in NPC and Neuron groups. Each group with *n* = 3 biologically independent samples, *p* value from the Empirical Bayes method in limma. Data were presented as mean values ± SD. Source data are provided as a Source Data file. **f** Immunostaining results showing Pum2 (red) and App (green) expression in the E14.5 dorsal forebrain. Hoechst (blue) indicates genomic DNA. The abbreviations VZ, IZ, and CP denote the ventricular zone, intermediate zone, and cortical plate, respectively. Scale bar, 100 μm. Representative images from two independent experiments.

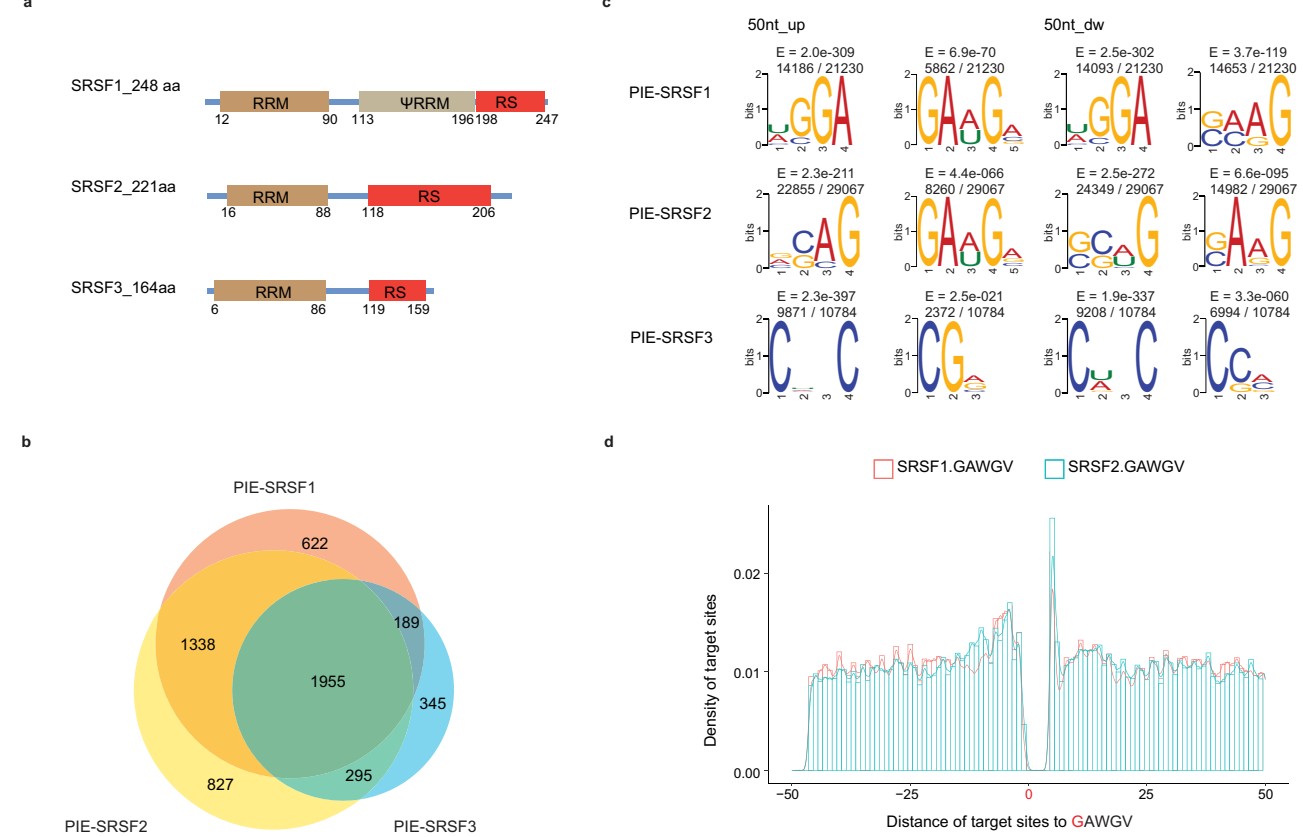

**Fig. 6 | PIE-Seq identifies SRSF1/2/3 target mRNAs. a** Schematics of SRSF1, SRSF2 and SRSF3 protein domains. **b** Venn diagrams showing the comparison of target genes among PIE-SRSF1, PIE-SRSF2 and PIE-SRSF3. **c** The top two enriched motifs within the ±50-nt region of PIE-SRSF1, PIE-SRSF2 and PIE-SRSF3 target sites. **d** Distribution of PIE-SRSF1/2 target sites flanking the GAWGV motif (the first Guanine in the motif was designated as position 0).

(Supplementary Fig. 8a)[58,59], nominating 775 genes as previously unreported NOVA1 targets (Supplementary Fig. 8a).

DART-Seq has been used to identify target genes for YTH proteins[24] and we seek to determine whether PIE-Seq achieves comparable results. The YTH family proteins function as direct m[6]A readers for post-transcriptional gene regulation and play essential roles in neural development[60,61]. In non-targeting APAD controls, we found that Apobec1 preferably edited the C in the ACH sequences that highly overlapped with the m[6]A DRACH motif (Supplementary Fig. 3a and Supplementary Fig. 7c). Analyzing sequences upstream and downstream of the edited sites avoided the interference of such reminiscent binding preferences, but the strategy split the DRACH sequences in the PIE-YTH cases which contain the edited A or C base (Supplementary Fig. 7d), depleting the complete DRACH motif. Consequently, the motif analyses in all four PIE-YTH RBPs identified GC-rich sequences (Fig. 7c). Despite this limitation, the target genes identified by PIE-YTHDF1 and PIE-YTHDF2 were significantly confirmed by reported PAR-CLIP target genes (Supplementary Fig. 8b, c)[62,63]. These results indicate that PIE-Seq identifies target genes of YTH proteins and is limited in motif discovery when the binding sites are edited.

We next focused on target gene discovery by PIE-Seq (Fig. 7b). We compared PIE-Seq results to previous studies available for 19 RBPs, and on average, 63% of PIE-Seq target genes were cross-validated by CLIP- or RIP-seq methods (Supplementary Fig. 8d). Taking PUM2 and NOVA1 as examples, 69% of target genes in PIE-PUM2 and 61% in PIE-NOVA1 were cross-validated by previous reports (Fig. 2k and Supplementary Fig. 8a, d). Furthermore, homologous RBPs tend to share overlapping target genes. For example, 87% of PIE-LIN28B target genes were uncovered by PIE-LIN28A (Supplementary Fig. 8f), and over 70% of

targets were shared between any two RBPs for CELF1/2/4 or KHDRBS1/2/3 family members (Supplementary Fig. 8g, h).

PIE-Seq analyses using the dual-deaminase strategy led to more biological insights and better target discovery than individual editors for PUM2 and additional RBPs such as FMR1 and YTHDF1/2 (Fig. 2g and Supplementary Data 2). For example, FMR1 is associated with the Fragile X syndrome (FXS) and has been reported to regulate protein translation through binding to the coding regions without consensus sequence preferences[64,65]. PIE-FMR1 identified UAUWW as the most enriched motifs flanking C-to-U target sites, while distinct GC-rich motifs were enriched surrounding A-to-I target sites (Fig. 7c and Supplementary Data 3). We uncovered 956 target genes using PIE-FMR1, including 304 target genes not reported in FMR1 PAR-CLIP[64] (Supplementary Fig. 8i). PIE-FMR1 uncovered 56 genes that have been associated with autism spectrum disorders (501 gene, SFARI score <3 and detectable in HEK293FT cells, Supplementary Fig. 8i). These results indicate that dual RNA deaminases in PIE-Seq detected different FMR1 target mRNAs and binding contexts. Overall, enriched motifs were uncovered from both C-to-U and A-to-I editing sites for nine out of 25 tested PIE-RBPs. Importantly, C-to-U and A-to-I target sites uncovered different features for three RBPs (FMR1, KHDRBS2 and KHDRBS3) and comparable binding motifs for the other six RBPs (Supplementary Data 3). These results suggest that the dual-deaminase approach enhanced RBP target discovery when compared to individual editors.

Integrated target analyses of all 25 PIE-RBP showed that 56% (3462/7742) of target genes are regulated by six or more RBPs, and that 24% (1888/7742) of genes were targeted by one or two RBPs (Supplementary Fig. 8e). Each RBP or RBP family showed "signature" target genes. For example, the autism-associated *MED12* gene showed a unique PIE-FMR1 target site in the CDS (Fig. 7e). We also observed

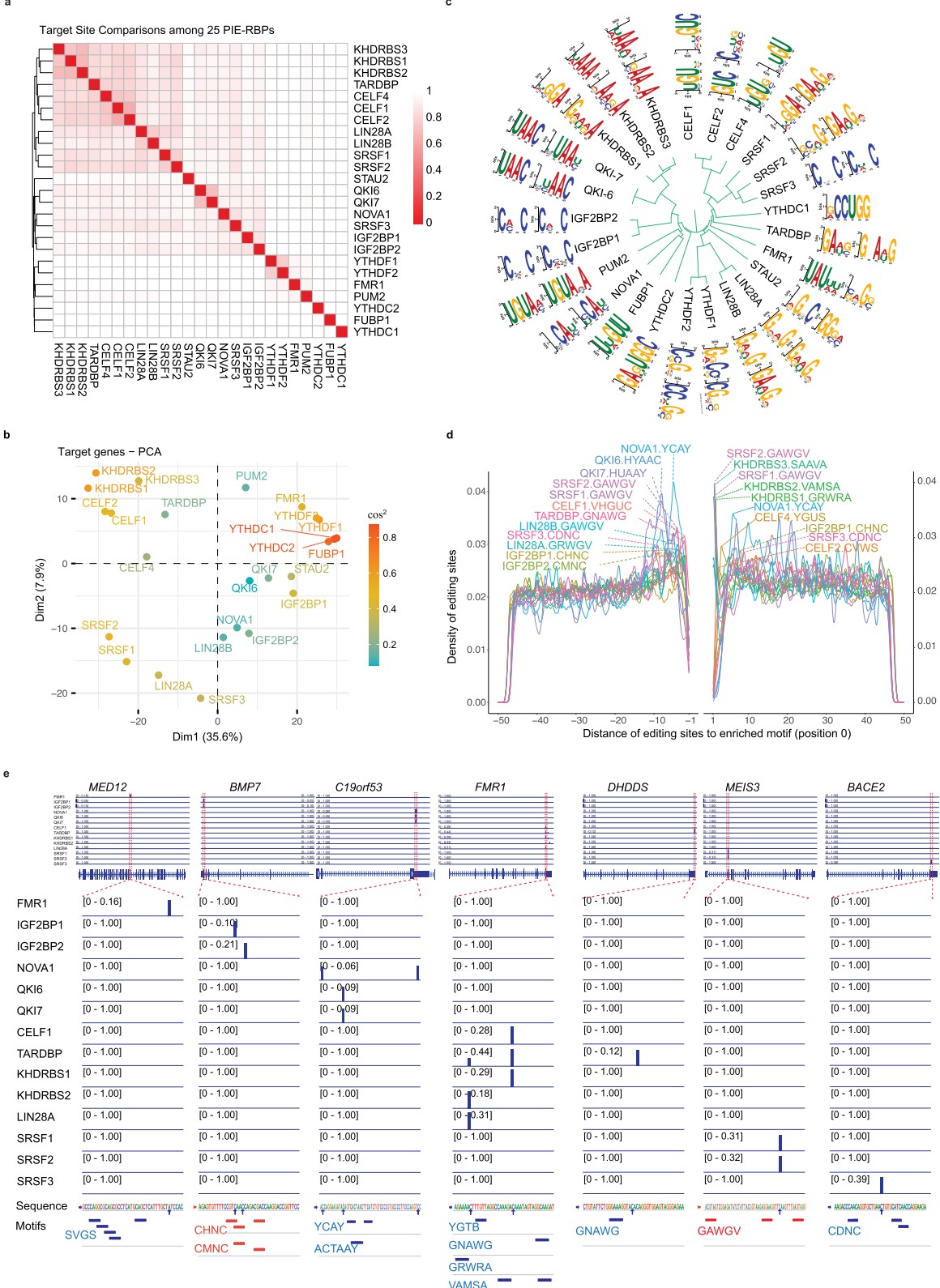

**Fig. 7 | PIE-Seq uncovers targets and binding features for 25 human RBPs.**
**a** Distance heatmap showing the comparisons of target sites among 25 PIE-RBPs. The color represents Euclidean distance (darker means closer). **b** PCA analysis of target genes identified by 25 PIE-RBPs. The colors indicate the squared cosine. **c** A phylogenetic tree drawn based on amino acid sequences showing the top sequence motifs of 25 PIE-RBPs. **d** The distribution pattern of target sites within ±50-nt distance to the representative binding motifs of indicated RBPs. The length of all motifs was scaled to one nucleotide and designated position 0. **e** Example target genes, target sites, and enriched motifs for PIE-RBPs. The vertical blue bars represent delta editing rates between PIE-RBPs to the APAD control.

genes that were targeted by a combination of RBPs. For example, the *FMR1* 3′ UTR showed target sites by CELF1, LIN28A, TARDBP, and KHDRBS1/2 (Fig. 7e). These results suggest that RBPs can work separately or in concert to regulate gene expression.

In summary, we developed and applied PIE-Seq to 25 RBPs. While a significant portion (63%) of target genes identified by PIE-RBPs are validated by CLIP or RIP datasets, on average, 37% of targets are unrevealed; we nominate more targets for RBPs such as TDP-43/TARDBP (1720 target genes), FMR1 (304), STAU2 (509), and LIN28A (2502) (Supplementary Fig. 8d). PIE-Seq uncovered canonical target genes and motifs for RBPs such as PUM2 and NOVA1, and revealed binding preferences for RBPs such as FUBP1 and KHDRBS proteins. Importantly, the edited sites are tightly associated with corresponding RBP binding motifs (Fig. 7d), suggesting that the deamination reactions introduced by PIE-Seq were restricted to local sequences nearby RBP binding sites. Thus, PIE-Seq takes advantage of dual deaminases and provides an orthogonal approach and resource to uncover RBP targets in mice and human cells.

## Discussion

Protein–RNA interaction plays central role in gene regulation. Here, we present the development and applications of PIE-Seq to identify RBP targets by dual C-to-U and A-to-I RNA editing. Integrating dual editors joined the advantages of Apobec1 and ADAR2dd, enhanced target detection, and permitted the direct comparison of their limitations. We applied PIE-Seq to 25 human RBPs and identified canonical and unreported binding features and target genes, such as bona fide binding motifs of PUM2 and NOVA1. We further showed that inducible, single-cell, and in vivo applications of PIE-Seq led to the versatile, sensitive and cell-type-specific discovery of RBP targets. PIE-Seq presents an orthogonal and widely applicable approach to investigating protein–RNA interaction.

PIE-Seq utilizes dual deaminases to reduce biases from RNA sequences and secondary structures. Both ADAR2dd and Apobec1 retained deaminase activities after fusing to 25 RBPs with flexible linkers. Several lines of evidence indicate that PIE-Seq is a robust approach to uncovering RBP targets: (1) PIE-MCP showed high on-site editing rates around MS2; (2) PIE-Seq uncovered canonical binding motifs for multiple RBPs such as PUM2, NOVA1 and SRSF1/2/3, and such motifs were consistently within the proximity of editing sites; (3) PIE-Seq target genes are highly concordant with CLIP-Seq, PAR-CLIP, RIP-Seq targets of the same RBPs, and have been further validated by RIP experiment; (4) PIE-Seq can be applied to intact tissues to identify cell-type-specific RBP targets; (5) Target sites, target genes, and enriched motifs show significant differences between RBP families, suggesting that RBPs are the main drivers for base editing; and (6) The majority of RBP isoforms and homologs display highly overlapping target genes and enriched binding motifs. These results indicate that PIE-Seq is robust for RBP target discovery and validation, serving as an orthogonal approach that complements CLIP-based methods.

Comparisons of C-to-U and A-to-I editing sites using PIE-Seq led us to find that Apobec1 consistently introduced more editing sites than ADAR2dd in both non-targeting controls and 25 RBP-fusion proteins (Supplementary Data 2). Overall, there are more genes showing C-to-U than A-to-I editing sites, and each gene, on average, shows more C-to-U sites than A-to-I. While thorough RNA editing is advantageous to uncover all possible RBP targets, the high stochastic editing activity of Apobec1 also introduced high background—highlighting the importance of parallel non-targeting controls. We have compared single APAD control with the combination of three controls (APADcom), and concluded that a single APAD control is sufficient for target discovery (Supplementary Fig. 6b). In contrast, ADAR2dd showed minimal background editing, and most genes had one A-to-I editing site if there was any. As such, C-to-U editing sites are more consistent than A-to-I in uncovering target genes and RBP motifs, while A-to-I editing uncovers

unique features/motifs for RBPs such as FMR1. The deaminase activities observed here in mammalian cells appear different from findings in Drosophila[66]. It is worth mentioning that C-to-U and A-to-I editing sites may call out the same target gene, but this is not always true due to imbalanced deaminase activities and sequence preferences. These results suggest that combining dual editors enhances RBP target detection.

Through comparing PIE-Seq results to previously reported studies, we found PIE-Seq reproducibly identified known RBP targets and canonical binding features, and also significantly expanded the target gene reservoir for the majority of RBPs studied here. For example, PIE-Seq nominated 612 additional target genes for PUM2 and 1720 target genes for TARDBP. RIP-qPCR validated more than half of the tested PIE-PUM2 target genes (Fig. 3f). Noticeably, RIP-seq and different types of CLIP-Seq datasets displayed significant variations in terms of RBP target genes, such as the cases for PUM2 and NOVA1 (Fig. 2k and Supplementary Fig. 8a). Thus, PIE-Seq provides an orthogonal and antibody-free approach to identifying and validating RBP targets.

PIE-Seq identified previously unknown RNA-binding motifs for several RBPs. For example, PIE-Seq identified an AG-rich motif surrounding C-to-U target sites for three KHDRBS proteins, and this motif is also distinct from motifs of other RBPs identified in this study. For another example, the UWGUU motif identified in PIE-FUBP1 was supported by a previous report that FUBP1 promotes the inclusion of DMD exon 39 through binding to UGU sequence in intron 38[67]. Very interestingly, even though the enriched motifs for different RBPs are diverse, most motifs tend to be closely upstream or downstream of the corresponding editing sites, suggesting that PIE-Seq target sites can be used to track RBP binding footprint. These results indicate that PIE-Seq robustly identified RBP binding sites.

Compared with CLIP-Seq-based methods[17,68], PIE-Seq has the following advantages: (1) PIE-Seq is convenient to implement because it involves transient protein expression and RNA sequencing, eliminating the requirement of UV crosslinking, immunoprecipitation, or other CLIP-Seq-specific experimental steps/reagents; (2) PIE-Seq constructs could be expressed in vivo and identify endogenous targets from cell types in animal tissues such as the embryonic brain; (3) PIE-Seq is applicable to a very small amount of input when combined with concurrent single-cell RNA-Seq methods. Thus, PIE-Seq makes it feasible to analyze protein–RNA interaction in vivo and among different cell types.

## Limitations

The nature of C-to-U and A-to-I base editing brings in intrinsic preferences for Cs and As at the editing sites. We noticed that sequences flanking (±4-nt) C-to-U and A-to-I editing sites in APAD and PIE-RBPs displayed reminiscent binding motifs of Apobec1 and ADAR2dd (Supplementary Fig. 3a), which would introduce bias for motif analysis especially when the editing sites were part of the binding motifs. For example, Apobec1 tends to edit the C base in ACH motifs (Supplementary Fig. 7d), which highly overlaps with the DRACH motif for m6A binding proteins. Apobec1 has been fused with the YTH domain of YTHDF2 to identify m6A RNA methylation sites in cultured cells[24,69], and here our results indicate that extra cautions are needed when the binding motif resembles or includes the deaminase residual sequence. In addition, PIE-Seq uncovered fewer targets than other methods for the YTH proteins, possibly due to the overlapping ACH motif, incompatible protein conformations, or YTHDF-mediated stress granule formation[70], which may impair deaminase activity. Except for PUM2, PIE-Seq was performed here mostly in HEK293FT cells, which may not be the best model for neurological disorders. To minimize the residue sequence preferences of RNA deaminases and overcome the lower editing efficiency of ADAR2dd, we have screened other deaminase modules, including TadA*[71], which has a distinct molecular origin from ADAR2dd or Apobec1. We have shown that TadA* had strong

editing activities in the MCP-MS2 system but led to limited editing sites when benchmarked in cultured cells. Despite that, the majority of PIE^TadA*-PUM2 genes were reproduced in other PIE-PUM2 experiments (Fig. 2k and Supplementary Fig. 2f), suggesting that TadA* variants can be further optimized for PIE-Seq. These intrinsic C- and A-base preferences of single-base editors and the imbalanced editing activities between Apobec1 and ADAR2 highlight the importance of joining dual deaminases to enhance target discovery.

In summary, the PIE-Seq dual-deaminase method is robust to investigate protein–RNA interaction. PIE-Seq significantly expanded the spectrum of target genes and binding preferences for 25 RNA-binding proteins. PIE-Seq is easy to implement, free from special equipment or reagents, sensitive enough for single-cell inputs, and applicable for dissecting protein–RNA interaction in the developing mouse brain. PIE-Seq has the potential to uncover RNA regulatory mechanisms among cell types in intact tissues.

## Methods

### Ethical statement
Our research complies with all relevant ethical regulations, and the animal protocol has been reviewed and approved by the Institutional Animal Care and Use Committee (IACUC) of the University of Chicago (Approval number: 72543).

### Cell culture and transfection
HEK293FT cells (Fisher Scientific, Catalog number R70007) were cultured at 37 °C in DMEM (Gibco cat. No. 10566024) supplemented with 10% fetal bovine serum (Gibco cat. No. 26140079) in a humidified incubator with 5% carbon dioxide. For each well in a 12-well plate (Fisher Scientific cat. No. 087723 A), $8 \times 10^5$ cells were transfected in suspension with lipofectamine 2000 (Fisher Scientific cat. No. 11668019) and changed to fresh medium 4 h after transfection. The cell line was validated for mycoplasma negative. More cell culture and transfection details are seen in Supplementary Note 2.

### In utero electroporation (IUE) in mice
CD-1 mice were purchased from Charles River, housed and bred at the University of Chicago Animal Care Facility under a 12-h dark/12-h light cycle and ambient temperature. The day (noon) when a vaginal plug showed, was designated E0.5. IUE procedures were performed following previously described protocols[72] and was approved by the IACUC of the University of Chicago (Approval number: 72543). Briefly, DNA plasmids were prepared using an endotoxin-free isolation kit and diluted in DNase/RNase-free water. The surgery was performed on a pregnant mouse under aseptic conditions and anesthesia, achieved by introducing 3% isoflurane at 0.8 L/min oxygen until the mouse was asleep and maintaining 1.5% isoflurane during the procedure. A proper concentration (about 1 μg/μl) of DNA solution with 0.05% Fast Green dye was injected into the E13.5 brain lateral ventricle in utero via glass micropipettes, and five square fixed-potential pulses with a duration of 50 ms per pulse were administered by an Electro Square Porator (ECM830, BTX). After the surgery, the uterus was returned, and the wound was closed. The mouse received postoperative care, including monitoring, analgesics and ensuring proper healing.

### Plasmid construction
pR059_pCAG-*Apobec1*: *rApobec1* was amplified from pCMV-BE3 (Addgene #73021) with primers XR001/XR189 and inserted into the pCAGIG backbone.

pR060_pCAG-*MCP-Apobec1*: *MCP* sequence was amplified from *MS2_GFP* (Addgene #61764) with primers XR185/XR186, *SGGS-XTEN-SGGS* linker was amplified from pCMV_ABEmax (Addgene #112095) with primers XR187/XR188, and these two PCR fragments were assembled with *rApobec1* into pCAGIG with Gibson Assembly (NEB, the same method was used below if not described otherwise).

pR087_pCAG-*mCherry-2XMS2*: 2xMS2 sequence was amplified from *a 24xMS2* reporter plasmid (Addgene #61762) with primers XR308/XR309 and inserted into the 3′ UTR of *mCherry* in the pCAG-*mCherry* backbone.

pR088_pCAG-*TadA-TadA*-dPspCas13b-Apobec1*: *TadA-GS-XTEN-GS-TadA*-GS-XTEN-GS* was amplified from pCMV-ABEmax (Addgene #112095) with XR147/XR144, dPspCas13b was amplified from pC0055-CMV-dPspCas13b-GS-ADAR2DD(E488QT375G)-delta-984-1090 (Addgene #103871), *GS-XTEN-GS-Apobec1* from pR060 with XR312/XR313, and these PCR fragments were assembled into the pCAGIG backbone.

pR090_pCAG-*TadA-TadA*-MCP*: The *TadA-GS-XTEN-GS-TadA*-GS-XTEN-GS* from pCMV-ABEmax (Addgene #112095) was amplified with XR318 and XR319, and the *MCP* from MS2_GFP (Addgene #61764) was amplified with XR320 and XR321; these two fragments were assembled into the pCAGIG backbone.

pR100_pCAG-*TadA*-MCP*: *TadA*-MCP* from pR090 was amplified with XR356 and XR357 and inserted into the pCAGIG backbone.

pR102_pCAG-*TadA*(V82G)-MCP*: *TadA*(V82G)-MCP* from pR090 was amplified with XR356/360 and XR361/357; these two fragments were assembled into the pCAGIG backbone.

pR163_pCAG-*MCP-ADAR2dd*: hADAR2dd was PCR amplified from pC0055 with primers XR591/XR592, cut with *Bst*E2 and *Not*1, and replaced *rApobec1* CDS in the *MCP-Apobec1* plasmid.

pR284WT_pCAG-*MCP-APOBEC3A*: APOBEC3A was amplified from A3Ai-Cas9n-UGI-NLS (Addgene #109425) with XR892 and XR893 and then assembled into the pCAGIG backbone.

pR284Mut_pCAG-*MCP-APOBEC3A(K30R/Y132G)*: APOBEC3A(Y132G/K30R) was amplified from A3Ai-Cas9n-UGI-NLS (Addgene #109425) by three segments: A3A_part1 with XR892/XR888, A3A_part2 with XR889/XR890, A3A_part3 with XR891/XR893, and these three fragments were assembled into the pCAGIG backbone.

pR008_APAD: *rApobec1*-XTEN was amplified from pCMV-BE3 (Addgene #73021) with primers XR001/XR024, and hADAR2dd was amplified from pC0055 (Addgene #103871) with primers XR025/XR006. The two PCR fragments were assembled into pCAGIG.

pR023_PIE-*PUM2* and pR018_PIE-*FMR1*: *PUM2* CDS was amplified from pFRT/FLAG/HA-DEST *PUM2* (Addgene #40292) with primers XR058/XR059, and *FMR1* CDS was amplified from pFRT-TODest*FLAGHAhFMRPiso1* (Addgene #48690) with primers XR044/XR045, and these CDS sequence were inserted between *rApobec1*-XTEN and *GS-linker-ADAR2dd* in pCAGIG backbone with T4 ligation, respectively.

pR203_AP-AD and additional PIE-RBPs: rApobec1-XTEN was amplified from pCMV-BE3 (Addgene #73021), and SGGS-XTEN-SGGS linker and hADAR2dd was amplified from pC0055 (Addgene #103871). The three PCR fragments above were assembled into pCAGIG with Gibson Assembly (NEB) to get pR203. The CDS of mCherry, dCas13b and more PIE-RBPs were inserted between Apobec1 and hADAR2dd through Gibson Assembly.

pINDUCER21_APAD and pINDUCER21_PIE-PUM2 were constructed by Gateway LR cloning with pINDUCER21 (ORF-EG) backbone (Addgene #46948).

Primer sequences are listed in Supplementary Data 1.

### Immunofluorescence staining
Immunofluorescence staining was done 24 or 48 h after transfection or electroporation. Cells were rinsed once with 1x PBS and fixed with 4% PFA at room temperature for 15 min. Fixed cells or brain slices were rinsed with 1x PBS, incubated with blocking buffer (1x PBS with 0.3% Triton-X-100 and 5% donkey serum) at room temperature for 30 min, and further incubated with primary antibodies diluted in blocking buffer overnight at 4 °C. After three brief rinses in 1x PBS, slides were incubated for 1 h at room temperature with fluorophore-conjugated secondary antibodies (Donkey anti-

chicken 488 (Jackson ImmunoReseach, 703-546-155), donkey anti-mouse 488 (Thermo Scientific, A21202), donkey anti-rabbit 594 (Thermo Scientific, A21207), donkey anti-rat 647 (Thermo Scientific, A48272); 1:1000 dilution for all). Slides were scanned with a Leica SP5 inverted confocal microscope. The following primary antibodies were used: anti-HA (Millipore 3F10 clone, rat, 1:1000), anti-GFP (Abcam ab13970, chick, 1:2000), anti-PUM2 (Bethyl A300-202A, rabbit, 1:1000), anti-CD24-PE (BioLegend 138504, rat, 1:1000), anti-CD133-APC (BioLegend 141208, rat, 1:1000), anti-PAX6 (Covance PRB-278P, rabbit, 1:500), anti-APP (BioLegend 802803, mouse, 1:1000).

## Western blot

Cells from each group were lysed with RIPA buffer (Thermo Fisher, PI89901) containing 1x cOmplete EDTA-free protease inhibitor cocktail (Sigma-Aldrich 11836170001) and shaken on ice for 20 min. The lysates were centrifuged, and the supernatants were denatured in 1x loading buffer at 95 °C for 10 min. Protein samples were then loaded and separated on SDS-PAGE gel in Tris/glycine/SDS buffer, transferred onto PVDF membrane, and incubated overnight at 4 °C with primary antibodies: anti-PUM2 (Bethyl A300-202A, rabbit, 1:5000) and anti-LMNB1 (Santa Cruz sc-6217, goat, 1:3000. Secondary antibodies donkey anti-rabbit IgG (LI-COR Biosciences, 926-32213) and donkey anti-goat IgG (LI-COR Biosciences, 925-68074) were used for detecting PUM2 and LMNB1 proteins, and the membranes were scanned with a LI-COR Odyssey DLx imager. Images were processed and quantified with Adobe Photoshop.

## cDNA synthesis and Sanger sequencing

Total RNA was extracted from transfected cells with a Direct-zol RNA microprep kit (ZYMO R2061), and reverse transcribed to first-stranded cDNA with SuperScript IV Reverse Transcriptase (Thermo Fisher 18090050) following manufacturer's instructions. PCR products were purified, and Sanger sequenced.

## Fluorescence-activated cell sorting (FACS)

HEK293FT cells were washed with 1x DPBS without Calcium or Magnesium (Gibco 14190250) 24 or 48 h after transfection and digested with 0.25% Trypsin-EDTA (Gibco 25200114). Cells were diluted with 1x DPBS supplemented with 10% FBS and filtered through 70-μm cell strainers (Fisherbrand 22363548). IUE brains were dissected in cold 1x DPBS, digested with papain (Worthington Biochemical LK003150) for 15 min at a 37-degree incubator, and then resuspended in 1x DPBS with 1% BSA, filtered through 100-μm cell strainers (Fisherbrand 22363549). Flow cytometry was performed on an ArialIIu with the FACSDiva operating system (BD Biosciences. Cells expressing control vectors or PIE-RBPs were sorted for the top 40% of gated cells with the FITC or APC/PE signals into pre-chilled Trizol.

## High-throughput sequencing

Bulk sequencing libraries were prepared with TruSeq Stranded mRNA Library Prep kit (Illumina 20020594), except that NOVA1, FUBP1, TARDBP, CELF1/2/4, and KHDRBS1/2/3 were prepared with Stranded Total RNA Prep Ligation with Ribo-Zero Plus kit (Illumina 20040525). Each sample was barcoded using IDT for Illumina-TruSeq RNA UD Indexes (IDT 20040553). Libraries were sequenced on the Illumina Next-SEQ500 with 75 bp pair-end runs or Nova-Seq with 60 bp pair-end runs by the University of Chicago Genomics facility.

## Editing sites analysis

Following the RNA variant calling pipeline with JACUSA2 package[38], raw Fastq files were trimmed, aligned and mapped to hg38 or mm10 with STAR[73]. Duplicate reads were identified with the MarkDuplicates tool in Picard (v2.19.1), and C-to-U and A-to-I editing variants were called with JACUSA2 (v2.0.0-RC23) with at least 5 reads covering each

editing site as a cutoff. Known SNPs from human dbSNP version 138 or mouse dbSNP database (for the mouse brain study) and empty control sites were filtered out. The RBP target sites (C-to-T or A-to-G) were further analyzed with JACUSA2helper-1.99-9200 in R 4.2.0. The RBP target sites were called with the following filtering criteria: (1) For cultured cells, the editing rate was no less than 5% in PIE-RBP and at least twofold of that in APAD or APADcom control, log-likelihood $z$-score ≥4; (2) For in vivo mouse brain cells, the editing rate was no less than 5% in PIE-RBP, and at least 1.5-fold of that in deaminase control APAD, log-likelihood $z$-score >1. The genes with target sites were considered as RBP target genes. The control sites in APAD and APADcom were generated by comparison to empty control using the same pipeline. Detailed analysis pipeline in Supplementary Note 2.

## Differential gene expression (DGE) analysis

DGE analysis from RNA-seq data followed the fastp (v0.23.2), STAR (v2.6.1b) and DESeq2 (v3.12) (human samples), or Rsubread (v3.14) and limma (v3.42.1) (mouse samples) workflows[73–76]. Reads were aligned to the customized GRCh38 genome that included transgene sequences. Raw counts were transformed into TPM values for comparison between groups.

## Genomic distribution, motif analysis and phylogenetic tree plot

Genomic distribution of control sites in APAD or APADcom and target sites in PIE-RBPs was analyzed with "bed2annotation.pl" command in the CTK pipeline[77]. Enriched motifs for PIE-RBPs were identified by DREME (v5.4.1)[78] in sequences spanning ±50-nt of target sites, with the sequences ±50-nt of control sites in APAD or APADcom group as background. The deaminases editing sequence motifs (±4-nt flanking editing sites) were generated with HOMER (v4.11)[79]. The phylogenetic tree for RBP proteins was plotted with UPGMA (unweighted pair group method with arithmetic mean) method in MEGA11[80].

## CLIP data analysis

Published CLIP data: PUM2 PAR-CLIP raw peak list[15] was reannotated with hg38 and then ranked by frequency of a crosslinking signal. RNA-seq data for QKI-6, IGF2BP1, IGF2BP2 PAR-CLIP from GEO accession code "GSE21578"[15], RNA-seq data for FMR1 PAR-CLIP from GEO accession code "GSE39686"[64], NOVA1 eCLIP RNA-seq data from BioProject under accession code "PRJNA670687"[59] and mouse Nova1 HITS-CLIP RNA-seq data from GEO accession code "GSE69711"[58] have been reanalyzed using the CTK pipeline[77]. CLIP peaks were called with statistical significance (FDR <0.05), and genes with significant CLIP peaks were considered RBP targets.

## RNA immunoprecipitation and RT-qPCR validation

We ranked the PIE-PUM2 target genes by calculating the sum of editing scores for each target site using the JACUSA package. Based on this ranking, 42 genes that emerged as the top-ranked targets were selected for validation with RIP-qPCR. For RIP, we used the Magna RIP RNA-Binding Protein Immunoprecipitation Kit (EMD Millipore 17-700) and followed the manufacturer's protocol. Briefly, $2 \times 10^7$ HEK293FT cells were lysed in 100 μL of RIP Lysis Buffer (with protease inhibitors and the RNase inhibitor). About 5 μg of PUM2 and control IgG antibody were incubated with protein A/G magnetic beads for 30 min at room temperature with rotation. The lysates were incubated with the antibody-bead complex for overnight at 4 °C with rotation. We used a magnet to immobilize the beads, discarded the supernatant, washed the beads multiple times, and extracted RNA from the purified complexes. Then we analyzed targets with qRT-PCR. Sequences for RT-qPCR primers are listed in Supplementary Data 1.

## Statistics and reproducibility

No data were excluded from the analyses; The Investigators were not blinded to allocation during experiments and outcome assessment. All

quantitative data were presented as the mean ± SD, as indicated by at least three independent experiments or biological replicates unless otherwise stated. The statistical tests and *p* values were described in the figure legend for each experiment. All data shown are representative of two or more independent experiments with similar results, unless indicated otherwise.

## Reporting summary

Further information on research design is available in the Nature Portfolio Reporting Summary linked to this article.

## Data availability

Data supporting the findings of this work are available within the paper and the Supplementary Information. Source data are provided with this paper. The raw and processed sequencing data generated in this study have been submitted to the NCBI Gene Expression Omnibus under accession code "GSE155844". Data used in this work includes RNA-seq data for QKI-6, IGF2BP1, IGF2BP2 PAR-CLIP from GEO under accession code "GSE21578"; RNA-seq data for FMR1 PAR-CLIP from GEO under accession code "GSE39686"; NOVA1 eCLIP RNA-seq data from BioProject under accession code "PRJNA670687" and mouse Nova1 HITS-CLIP RNA-seq data from GEO under accession code "GSE69711". Source data are provided with this paper.

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

## Acknowledgements

The authors would like to thank Professors Marcelo Nobrega, Matthew Stephens, Chuan He, Tao Pan, Sangram S. Sisodia and Paschalis Kratsios for advice and reagents on this study; thank Dongyue Xie, Bowei Kang and Ankeeta Shah for their assistance on data analyses; thank Pieter W. Faber and the Genomics Facility at UChicago for DNA sequencing; thank the Cytometry and Antibody Technology Facility at UChicago for cell sorting; and thank Peter Carbonetto and the Research Computing Center for technical assistance. This study is supported by K01-MH109747 (NIMH) and DP2-GM137423 (NIGMS) to X.Z.

## Author contributions

X.Z. conceived, initiated and supervised this study. X.R. performed the experiments, collected and analyzed the data. K.H. analyzed data. X.R. and X.Z. interpreted the data and wrote the manuscript.

## Competing interests

The authors declare no competing interests.

 
