## [Peer Review File · Nature Communications]

REVIEWER COMMENTS

Reviewer #1 (Remarks to the Author):

Summary

Zhang and colleagues combined two existing technologies to identify higher-confidence RNA-binding sites for RBPs. The authors went through many permutations of editing enzymes, and arrived upon the final PIE-seq construct: Apobec1-RBP-Adar2dd. The authors made good use of using the non-targeting APAD negative control for measuring background editing for calculating significance. They show that this model system can work under inducible promoter design as well as in single cells (by SMRT-seq). They also showed that this system operates as expected and can reveal cell-type specific RBP binding in vivo through mouse-brain electroporation in utero. Finally, the authors expand their manuscript by characterizing a total of 25 RBPs using their PIE-seq approach in HEK293FT cells. Overall, the experiments were carried out well, and show that the proposed approach is effective. On the other hand the model based on transient or inducible overexpression, above physiological levels, dampens the enthusiasm for the method. The manuscript may also benefit from editing for smoothness.

There are also several areas that will require addressing prior to publication.

Major points:

1. The most striking aspect of this manuscript was the lack of concordance in results across the PIE, RIP, and CLIP approaches (Fig. S2F,G). PIE-PUM2 was only showed in comparison to CLIP (Fig. S2F) and separately compared to RIP (Fig. S2G). While it is indeed clear that there is some overlap in these pairwise comparisons, it would be better to understand the full picture of how each method compares to each other, including between CLIP and RIP approaches. I'd suggest using something like an UpSetR plot (<https://github.com/hms-dbmi/UpSetR>) that can show an intuitive way of looking at an n-dimensional Venn-diagram like plot. This will allow the reader to understand all of the comparisons that do and don't replicate across different technologies. A similar plot would be useful for Fig. 2G-I, and in several other instances where Venn-Diagrams were used.
2. In Figure 4D, the authors perform RIP PCR on a TOP2A target locus. IgG negative control is shown, but is also used as the normalization factor. Log₂(FC) is shown relative to IgG, but it looks like each measure for PUM2 was paired to a single IgG measure. This artificially shrinks the variance of the negative control to zero; an alternative that would allow the negative control to have its variance measured as well is to examine the log₂(FC), this is required for proper statistical comparison between IgG and anti-PUM2. Additionally, the y-axis title is different between Fig. S3D and Fig. 3D (which is noted as the same data in the legend of Fig. S3D); however, in Fig. 3D, this is shown as logFC from input, rather than logFC from IgG negative control (which I think may be the correct label). This should updated for clarity. Additionally the results of Fig. S3D are discussed (line 204) prior to the experimental design (Inducible PIE-Seq section), which made it a bit difficult to understand when read for the first time. In Fig. S3D it's also unclear why these 22 targets of the 41 discussed (line 202). Were these arbitrary selections, or was

there a selection criterion used? Fig. S3D may also benefit from sorting the targets left to right based on average logFC; this would likely make more clear the pattern that the higher logFC targets were those that replicated in all techniques. It's also not clear in Fig. S3D if the Tet-on annotation is referring to Tet-low or Tet-high.

3. The Tet inducible system is nice, because it can more closely mirror endogenous expression levels, and may show more natural targets compared to over-expression. However, to show this, in Fig. S3E, the authors look at counts per million (presumably from RNAseq, although from the text, figure, and legend it's hard to say). In this panel, expression of PUM2 is shown, but a much more useful comparison would be endogenous vs exogenous PUM2 (both at the RNA and protein level). To really do this comparison however, a custom reference genome would be required that includes all of the used transgenes. Without this, it's difficult to impossible to know how closely the dox conditions match endogenous levels because reads from PIE-PUM2 will be added on top of reads from the endogenous locus. The authors should both clarify the source material and experimental design (presumed RNAseq, the host cells, etc), and if it is indeed RNAseq data, re-align and quantify with a custom reference, in each condition showing endogenous and exogenous CPM. In general, a model in which near-physiological levels are expressed would really increase the quality and reliability of the data.

4. It would be good to see a direct comparison between the motif enrichment in the transfected over-expression and the TET-inducible systems. However, it remains unclear what the impact of over-expression vs endogenous matched expression is. A useful analysis that could shed light on this is to classify edited sites as discovered or not discovered in OE/Tet-high/Tet-low, and perform motif analysis on sites that were discovered in all three, Tet-high and OE, or OE only, to directly quantify the impact of transgene over-expression on the net biological conclusion of a study.

5. HEK293FT cells, while a nice model system because for transfectability, are not a great system for testing the 21 RBPs associated with neuronal disorders. This is also the case for the FMR1 study in lines 364-378. This mismatch between disease relevant cell-types should at least be noted in the manuscript.

6. The authors state: " PIE-PUM2 uncovered a C-to-U target site on App 3' UTR nearby a PRE motif in NPC group but not in the neuron group (Fig. 5D), suggesting that PUM2 may inhibit APP translation in neural progenitors.

Is there any way to experimentally validate this interesting observation? Or has it been observed in CNS specific PUM2 KO animals?

Minor

1. In the first sentence, it should read "...are central to gene regulation." Without including the word "the" before gene regulation.

2. Musashi is spelled Masashi on page 2.

3. The difference between TadA and TadA* is somewhat unclear, although the variant V82G is noted later. Is this what TadA* indicates? It should be clearly defined at it's first mention.

4. Are the K30R/Y132G variants of APOBEC3A (line 79) catalytically dead? Their significance should be stated in the text.
5. The original XTEN paper should be cited on line 95.
6. RRM (RNA Recognition Motif) should be defined at its first use in the manuscript (line 288)
7. The APADcom control is slightly unclear, was this a triple transfection of all three negative control designs, or an in-silico average of the three taken separately? It would be better for the reader to change line 293 to say “the combined negative control” to make it clear the ADAPcom abbreviation indicates combined.

Reviewer #2 (Remarks to the Author):

In this manuscript, Ruan et al. constructed a dual-deaminase PIE-Seq method for identifying RNA binding protein (RBP) targets. The authors have benchmarked the method and demonstrated its sensitivity in single cells, the application in the developing brain, and the scalability with 25 human RBPs involved in diverse biological functions. Also, they applied PIE-Seq to single cells and cell types isolated from the developing mouse neocortex, further revealing the sensitivity and specificity of the method.

Overall, this is a well-structured manuscript with a refined workflow, aiming at more accurate identification of multiple RBP substrate information. However, despite its intriguing topic and clear data representation, there are still some limitations which detracts from its quality and novelty as the following concerns.

Major Points:

1. Most of the RNA deaminases present sequential or structural preference which would introduce sequencing bias. Considering that ADAR2 deaminase domain prefers to edit bulged A sites in double-stranded RNAs while Apobec proteins exhibit no double-stranded RNA preference, and only the RBP binding sites with double-stranded region on one side tend to be edited by ADAR2, the detected RBP binding sites flanked by A-to-I and C-to-U editing sites would probably be only a small percentage of the actual RBP binding sites, which may explain the relatively fewer target sites identified compared to PAR-CLIP-seq. Would it be possible to further optimize the deaminase selection or use engineered deaminase to prevent bias and increase the efficiency of identification?
2. Theoretically the target gene of RBP should contain both C-to-U and A-to-I target sites, since two editing proteins are fused at both ends of RBP. Why the identification criteria for target genes in this manuscript is that the gene contains any editing site? In addition, the authors excluded the edited base and explored the enriched RNA motifs 50-nt either upstream or downstream of the edited sites separately. Should the length be decided by the protein spatial size and linker length? How many bases

are there between different editing sites on the same gene? Is it possible to define RBPs targeted regions accurately to the area between two different editing sites?

3.As mentioned in the Method section, the author used 5% editing efficiency as a threshold to screen RBP target sites, which means that only 5 reads were edited when the site was covered by 100 reads. Is the threshold setting too low? Is the site coverage baseline considered, such as the minimum covered reads abundance of each site is 50? What is the basis of the screening thresholds for targeted sites set? How to exclude false positive or negative results? The authors should address the basis for these criteria.

4.For the cellular experiments, the authors mentioned the transfection time was 24 h or 48 h. Did the authors estimate the optimal transfection time? How to evaluate the transfection efficiency?

5.In lines 130-156, the author compared the editing efficiency of joint the two editing proteins and only single ones, but the results of Figs. 2G-I showed that the jointly discovered target genes were not very prominent. After deducting the background information of ADAP, the result of PIE-PUM2 seemed to be less noisy. Does AD-PUM2 or AP-PUM2 also consider deducting the background information using only AD or AP? What will be the results?

6.Compared with PAR-CLIP-Seq method, very few targeted genes in the YTHDF family were identified by PIE-Seq. Reasonable explanations should be given for this result.

Minor Points:

1.In lines 97-99, the authors mentioned two controls for PIE-seq, but in the following analysis, only ADAP control was considered. Whether the empty control is necessary or not should be addressed.

2.In lines 102-103, the transfected cells were performed RNA-Seq for polyA mRNAs. The main recognition and binding sites of some splicing proteins (such as SRSF proteins) are on pre-mRNAs. Whether PIE-seq should consider the method of rRNA deduction for sequencing or not?

3.When applying PIE-seq to single-cell sequencing, the 10-cell and single-cell samples showed higher C-to-U editing ratios than bulk samples, and more target genes were identified, which might be a rare case in single-cell sequencing. Possible validation assays for those identified target genes should be considered.

4.About the hyper-editing events which indicates 20 or more C-to-U editing sites per gene, is this standard defined by the authors or someone else? Is it necessary to normalize this standard by gene or mRNA length? This kind of information is missing in the manuscript.

5.The format of some pictures needs to be improved. For example, the dot diagram in Fig. 1B lacks the horizontal axis information, and label of y-axis should be percent (%); The legend needs a title to explain the specific meaning of the heat map in Fig.2J-K and Fig. 7A.

6.When comparing the results of the two treatment conditions, the significance test need to be mentioned, such as the result which is shown in Fig. S5D.

7. Generally, the presentation and clarity of some of the pictures should be improved to ensure that they are easy to be understood and aesthetically pleasing.

Reviewer #3 (Remarks to the Author):

In this manuscript, Ruan et al. report usage of PIE-Seq using dual deaminases in RNA binding proteins (RBP)-RNA interactions. Briefly, authors introduced A-to-I and C-to-U deaminase domains onto the RBPs of interest. Authors show reproducibility of this method in human HEK293FT cells, developing mouse cortex using in utero electroporation and an option to study 25 RBPs efficiently. Furthermore, authors developed inducible PIE-seq. Overall, this is a strong study with elegant experimental design and results that supports conclusions. This will be an important addition to the field of unbiased screens of RBP-RNA interactions. I have only minor suggestion:

- 1) Authors should include statistics for Figure 1C.

Key experiments/updates in the revised manuscript:

We highly appreciate the constructive suggestions from all reviewers. We have addressed the comments in the point-to-point responses (following pages) and updated the manuscript accordingly. The key experiments we have completed during the revision are:

- 1) Following Reviewer #1's suggestion, we have further characterized PIE-RBP expression.
 - a. We have built customized reference sequences for the transgenes and re-aligned RNA-Seq reads. We showed that the inducible PIE-PUM2 is closer to endogenous PUM2 than transient expression at both RNA (Fig. S3D) and protein (Fig. S3E) levels, while the transient expression system is adequate for uncovering RBP binding targets.
 - b. Single-cell and ten-cell samples showed higher target discovery rates (Fig. 4C and S4C) possibly due to higher PIE-PUM2 expression or higher RNA-Seq coverage. We have performed down-sampling analyses and showed that 25 million reads per sample are sufficient to capture over 65% of target genes (Fig. S4C-S4E).
- 2) Reviewer #2 raised questions about sequence biases from individual deaminases and the best way to call RBP binding sites and target genes. We have addressed these by:
 - a. Apobec1 and ADAR2 deaminase domains can introduce imbalanced editing sites and show limitations in motif discovery (Response and Fig. S7C-S7D). These observations indicate the importance of using dual deaminases shown in this study.
 - b. We have analyzed three aspects to address whether it is feasible to pinpoint RBP binding sites using PIE-Seq data: i) The MCP-MS2 reporter results (Fig. 1 and S1) indicate that the editing sites are close to the RBP binding site and are affected by the stem-loops. ii) Transcriptome-wide analyses of target sites showed that the editing happens within 20nt from the RBP binding sites (Fig. 3B and 7D). iii) The distance between the nearest C-to-U and A-to-I editing sites appears to peak at 60nt for 250 PUM2 target genes that carry both types of editing events (Response).
- 3) We have comprehensively compared the target genes identified by PIE-PUM2 to other technologies in an UpSetR plot (Fig. 2K), showing that most PIE-PUM2 target genes were re-discovered by AP-PUM2, PUM2-AD, PUM2 RIP-Seq or CLIP methods, and validated experimentally by RIP-qPCR (Fig. 3F).
- 4) We have provided further evidence that Pum2 binds to *App* 3'UTR and potentially attenuates its protein level (Fig. 5D-5E): we reanalyzed Pum2 CLIP data and found significant CLIP peaks in wild-type but not Pum2 knockout mice brains. We also identified complementary expression patterns of Pum2 and App proteins in the E14.5 neocortex.

We believe these new results significantly strengthened our initial conclusions and enhanced the impact of this work. In summary, PIE-Seq joins the advantages of single base editors and presents a robust new tool to investigate protein-RNA interaction. We respectfully submit the revised manuscript and the point-to-point responses for your consideration.

Point-to-point responses to REVIEWER COMMENTS

Reviewer #1 (Remarks to the Author):

Summary

Zhang and colleagues combined two existing technologies to identify higher-confidence RNA-binding sites for RBPs. The authors went through many permutations of editing enzymes, and arrived upon the final PIE-seq construct: Apobec1-RBP-Adar2dd. The authors made good use of using the non-targeting APAD negative control for measuring background editing for calculating significance. They show that this model system can work under inducible promoter design as well as in single cells (by SMRT-seq). They also showed that this system operates as expected and can reveal cell-type specific RBP binding in vivo through mouse-brain electroporation in utero. Finally, the authors expand their manuscript by characterizing a total of 25 RBPs using their PIE-seq approach in HEK293FT cells. Overall, the experiments were carried out well, and show that the proposed approach is effective. On the other hand the model based on transient or inducible overexpression, above physiological levels, dampens the enthusiasm for the method. The manuscript may also benefit from editing for smoothness. There are also several areas that will require addressing prior to publication.

Major points:

1. The most striking aspect of this manuscript was the lack of concordance in results across the PIE, RIP, and CLIP approaches (Fig. S2F,G). PIE-PUM2 was only showed in comparison to CLIP (Fig. S2F) and separately compared to RIP (Fig. S2G). While it is indeed clear that there is some overlap in these pairwise comparisons, it would be better to understand the full picture of how each method compares to each other, including between CLIP and RIP approaches. I'd suggest using something like an UpSetR plot (<https://github.com/hms-dbmi/UpSetR>) that can show an intuitive way of looking at an n-dimensional Venn-diagram like plot. This will allow the reader to understand all of the comparisons that do and don't replicate across different technologies. A similar plot would be useful for Fig. 2G-I, and in several other instances where Venn-Diagrams were used.

Thank you for the insightful comments and recommendations. We have applied UpSetR in the updated manuscript, and included an UpSetR plot (new Fig. 2K) to compare target genes among PIE-PUM2, AP-PUM2, PUM2-AD and PUM2 PAR-CLIP, RIP-Seq and HITS-CLIP in parallel to Fig. 2G-2I. Original results in Fig. S2F-S2G are now included in Fig. 2K.

2. In Figure 4D, the authors perform RIP PCR on a TOP2A target locus. IgG negative control is shown, but is also used as the normalization factor. Log₂(FC) is shown relative to IgG, but it looks like each measure for PUM2 was paired to a single IgG measure. This artificially shrinks the variance of the negative control to zero; an alternative that would allow the negative control to have its variance measured as well is to examine the log₂(FC), this is required for proper statistical comparison between IgG and anti-PUM2. Additionally, the y-axis title is different between Fig. S3D and Fig. 3D (which is noted as the same data in the legend of Fig. S3D); however, in Fig. 3D, this is shown as logFC from input, rather than logFC from IgG negative control (which I think may be the correct label). This should be updated for clarity.

Thank you for pointing this out. Now we have replaced this plot with the 42 RIP-qPCR results (previous Fig.S3D) and indicated standard deviations (Fig. 3F).

Additionally the results of Fig. S3D are discussed (line 204) prior to the experimental design (Inducible PIE-Seq section), which made it a bit difficult to understand when read for the first time.

We have reorganized the description and related discussion.

In Fig. S3D it's also unclear why these 22 targets of the 41 discussed (line 202). Were these arbitrary selections, or was there a selection criterion used?

These targets, including TOP2A and 41 others, have been ranked as the top 42 PUM2 targets based on the selection method where PIE-PUM2 target genes were ranked by their editing scores. We have now incorporated this selection criterion in the Methods section.

Fig. S3D may also benefit from sorting the targets left to right based on average logFC; this would likely make more clear the pattern that the higher logFC targets were those that replicated in all techniques. It's also not clear in Fig. S3D if the Tet-on annotation is referring to Tet-low or Tet-high.

We have replotted Figure S3D accordingly, now shown in Figure 3F. 27 out of 42 PIE-PUM2 target genes showed significant enrichment in the PUM2 RIP-qPCR when compared to the IgG control. "Tet-On" is the union of Tet-On.Low and Tet-On.High groups.

3. The Tet inducible system is nice, because it can more closely mirror endogenous expression levels, and may show more natural targets compared to over-expression. However, to show this, in Fig. S3E, the authors look at counts per million (presumably from RNAseq, although from the text, figure, and legend it's hard to say). In this panel, expression of PUM2 is shown, but a much more useful comparison would be endogenous vs exogenous PUM2 (both at the RNA and protein level). To really do this comparison however, a custom reference genome would be required that includes all of the used

transgenes. Without this, it's difficult to impossible to know how closely the dox conditions match endogenous levels because reads from PIE-PUM2 will be added on top of reads from the endogenous locus. The authors should both clarify the source material and experimental design (presumed RNAseq, the host cells, etc), and if it is indeed RNAseq data, re-align and quantify with a custom reference, in each condition showing endogenous and exogenous CPM. In general, a model in which near-physiological levels are expressed would really increase the quality and reliability of the data.

Following this constructive suggestion, we have re-aligned the reads with the hg38 + customized references for all these samples. Specifically:

- 1) We added APAD, PIE-PUM2(AP-PUM2-AD) transcript sequence into hg38 reference. We re-aligned and quantified TPM of the endogenous PUM2 using its 3'UTR and exogenous PIE-PUM2 (using rApobec1 in transient groups and Lentivirus-3'UTR in inducible groups as representative). The new results are presented in Fig. S3D, showing inducible fusion PUM2 RNA level is about 8 to 12 folds to endogenous PUM2 RNA. This comparison shows the inducible system generates closer PUM2 expression to its endogenous level.
- 2) We also included western blot results in Fig. S3E, showing that the fusion PIE-PUM2 protein level is about 6~11 folds higher in inducible groups than the endogenous PUM2.

4. It would be good to see a direct comparison between the motif enrichment in the transfected over-expression and the TET-inducible systems. However, it remains unclear what the impact of over-expression vs endogenous matched expression is. A useful analysis that could shed light on this is to classify edited sites as discovered or not discovered in OE/Tet-high/Tet-low, and perform motif analysis on sites that were discovered in all three, Tet-high and OE, or OE only, to directly quantify the impact of transgene over-expression on the net biological conclusion of a study.

Following this suggestion, we conducted motif analyses accordingly for the shared sites among all three groups, the common sites between Tet-high and OE, the sites shared by Tet-low and OE, and the OE-only sites. We successfully identified the core sequence of the PRE motif (UGUAYA or UGUAHA) in both Tet-high & OE sites and the OE-only sites, which have been updated in **Fig. S3F**. However, no enriched motifs were identified in the sites shared by the Tet-low group, possibly due to the limited number of sites. Therefore between Tet-high and OE, the motif enrichment was not sensitive to the level of transgene expression.

5. HEK293FT cells, while a nice model system because for transfectability, are not a great system for testing the 21 RBPs associated with neuronal disorders. This is also the case for the FMR1 study in lines 364-378. This mismatch between disease relevant cell-types should at least be noted in the manuscript.

We have added the discussion accordingly in the "Limitations" section.

6. The authors state: " PIE-PUM2 uncovered a C-to-U target site on App 3' UTR nearby a PRE motif in NPC group but not in the neuron group (Fig. 5D), suggesting that PUM2 may inhibit APP translation in neural progenitors. Is there any way to experimentally validate this interesting observation? Or has it been observed in CNS specific PUM2 KO animals?

1) We co-stained anti-App and anti-Pum2 in E14.5 mice brain slices (Fig. 5F) and found the complementary expression between Pum2 (primarily expressed in the ventricular zone) and App (high expression in the intermediate zone), suggesting that Pum2 may suppress App.

2) Reanalysis of reported Pum2 iCLIP-Seq data (GSE95197, Fig. 5D) showed significant Pum2 CLIP tags on the *App* 3'UTR in wild-type mouse brains but not in *Pum2* knockouts, suggesting that Pum2 binds to *App* 3'UTR. The bulk iCLIP experiments could not distinguish this binding specificity in neural cell types. We contacted the senior author about studying *Pum2* knockout mice or the brain samples and were told the mouse colony/sample was not available anymore.

Minor

1. In the first sentence, it should read "...are central to gene regulation." Without including the word "the" before gene regulation.

Thank you and we have corrected it accordingly.

2. Musashi is spelled Masashi on page 2.

We have corrected it accordingly.

3. The difference between TadA and TadA* is somewhat unclear, although the variant V82G is noted later. Is this what TadA* indicates? It should be clearly defined at its first mention.

The TadA* is evolved from wild type TadA, and the variant V82G was further engineered based on TadA*, named as TadA*(V82G). We have updated this in the manuscript.

4. Are the K30R/Y132G variants of APOBEC3A (line 79) catalytically dead? Their significance should be stated in the text.

The APOBEC3A K30R/Y132G variants were reported to have high deaminase activity on RNA substrates while no activity on DNA. However, these variants did not show RNA editing activity in the MS2-MCP system. We have added the original reference in the updated manuscript.

5. The original XTEN paper should be cited on line 95.

We have added the related reference accordingly.

6. RRM (RNA Recognition Motif) should be defined at its first use in the manuscript (line 288)

The RRM is defined at its first use.

7. The APADcom control is slightly unclear, was this a triple transfection of all three negative control designs, or an in-silico average of the three taken separately? It would be better for the reader to change line 293 to say "the combined negative control" to make it clear the ADAPcom abbreviation indicates combined.

The APADcom is an in-silico average of the data from three independent transfection groups (AP-AD, AP-dCas13b, and AP-mCherry-AD). We have updated the description accordingly.

Reviewer #2 (Remarks to the Author):

In this manuscript, Ruan et al. constructed a dual-deaminase PIE-Seq method for identifying RNA binding protein (RBP) targets. The authors have benchmarked the method and demonstrated its sensitivity in single cells, the application in the developing brain, and the scalability with 25 human RBPs involved in diverse biological functions. Also, they applied PIE-Seq to single cells and cell types isolated from the developing mouse neocortex, further revealing the sensitivity and specificity of the method.

Overall, this is a well-structured manuscript with a refined workflow, aiming at more accurate identification of multiple RBP substrate information. However, despite its intriguing topic and clear data representation, there are still some limitations which detracts from its quality and novelty as the following concerns.

Major Points:

1. Most of the RNA deaminases present sequential or structural preference which would introduce sequencing bias. Considering that ADAR2 deaminase domain prefers to edit bulged A sites in double-stranded RNAs while Apobec proteins exhibit no double-stranded RNA preference, and only the RBP binding sites with double-stranded region on one side tend to be edited by ADAR2, the detected RBP binding sites flanked by A-to-I and C-to-U editing sites would probably be only a small percentage of the actual RBP binding sites, which may explain the relatively fewer target sites identified compared to PAR-CLIP-seq. Would it be possible to further optimize the deaminase selection or use engineered deaminase to prevent bias and increase the efficiency of identification?

Thank you for the insightful comment! Addressing the sequencing bias of previously published methods such as TRIBE/ADAR2dd (deaminase domain, Cell, 2016; Science Advances, 2020) is exactly the motivation of this study. On the other hand, DART-Seq and STAMP/Apobec1 (Nature Methods, 2019, 2021) have their own limitations (see "Discussion - Limitations"). Our rationale is to expand the editable spaces using dual deaminase domains because ADAR2dd and Apobec1 show low/minimal RNA binding affinity without their RNA-binding domain or auxiliary proteins; and the RBP of interest will drive the binding to targets. Indeed, our results on RBP families indicate that RBPs play dominant roles in identifying target mRNAs (Figure 7).

To overcome the low editing efficiency of the ADAR2 deaminase domain, we have screened a number of engineered deaminases: TadA*(V82G) exhibited significantly higher A-to-I(G) editing efficiency than ADAR2dd in the MCP-MS2 assay (Fig. 1C). Surprisingly, TadA*(V82G) uncovered fewer A-to-I target sites when introduced into HEK293FT cells, probably due to toxicity (Figure 2H). We have been actively working on improving the system and further efforts are needed to optimize the deaminases.

2. Theoretically the target gene of RBP should contain both C-to-U and A-to-I target sites, since two editing proteins are fused at both ends of RBP. Why the identification criteria for target genes in this manuscript is that the gene contains any editing site?

We agree that RBP targets would ideally contain both C-to-U and A-to-I sites. Similar to previous TRIBE (A-to-I) reports, we uncovered much (order of magnitude) fewer A-to-I than C-to-U editing sites, probably due to the lower intrinsic editing efficiency of ADAR2dd. Consequently, there is a significant imbalance between the two types of editing sites. While this study reports genes that contain at least one significant editing site, it also allows the prioritization of targets with dual editing sites (Figure below).

In addition, the authors excluded the edited base and explored the enriched RNA motifs 50-nt either upstream or downstream of the edited sites separately. Should the length be decided by the protein spatial size and linker length?

The edited bases are strongly biased: they are always C for Apobec1, or A for ADAR2dd or TadA*. Thus, the edited base is excluded from motif analysis.

The 50-nt length was based on results from MS2-MCP (Fig. 1B, S1A-S1B), PUM2 (Fig. 3B), and the global profiling of editing sites (Fig. 7D). In all cases examined, the motif is highly enriched within 20nt from the editing sites.

How many bases are there between different editing sites on the same gene? Is it possible to define RBPs targeted regions accurately to the area between two different editing sites?

Thank you for this insightful question!

The short answers are: 1) The number of bases between different types of editing sites on the same gene are highly variable probably due to the imbalanced number of C-to-U and A-to-I sites (missing when compared to many C-to-U targets). 2) It is very challenging to define RBP targeted regions accurately to the area between two different (types of) editing sites, probably because of the imbalanced number of C-to-U and A-to-I sites and the flexibility of RNAs.

To answer this question, we first looked at the MS2-MCP experiment where the MCP binding site was fixed on the stem-loops. On the reporter construct, we observed that the A-to-I (G, ADARD2dd) editing site is at 26nt, and the C-to-U (Apobec1) editing site is at 73nt; both sites were 47 nt apart and in proximity to the well-defined RNA “stem-loops (38~56nt and 77~95nt)” (Fig. S1B). These observations suggest that the spacing of RNA editing sites was affected by both the RBP footprint and RNA secondary structures (MS2 stem-loops).

To further examine spacing between C-to-U and A-to-I(G) sites on the same gene, we examined the 250 PUM2 dual-edited target genes in Fig.2G to determine the nearest distances between A-to-I and C-to-U sites. The range of distances is [7nt, 7151nt], with a median of 393nt (Fig.A below). The distance between the nearest C-to-U and A-to-I sites appears to peak at 60nt. We speculate that the large range was caused by non-saturated and imbalanced editing and secondary RNA structures.

On the other hand, our results (Fig. 3B, Fig. 6D, and Fig. 7D) support that PIE-RBPs have a higher tendency to edit nucleotides in close proximity (within 20nt) to their binding regions/motifs. Moreover, PUM2 targets with both A-to-I and C-to-U editing sites showed higher enrichment of PRE motifs (described in the updated manuscript and Fig.B below). However, we acknowledge that it is challenging to define RBP target regions accurately to the area between two different types of editing sites.

3.As mentioned in the Method section, the author used 5% editing efficiency as a threshold to screen RBP target sites, which means that only 5 reads were edited when the site was covered by 100 reads. Is the threshold setting too low? Is the site coverage baseline considered, such as the minimum covered reads abundance of each site is 50? What is the basis of the screening thresholds for targeted sites set? How to exclude false positive or negative results? The authors should address the basis for these criteria.

Thank you for the constructive comments! Our overall sequencing read coverage is 60X, and the 5% threshold would require 3 or more reads, which is on par or more stringent than commonly used threshold (2 reads). On top of this coverage threshold, we also performed stringent statistical testing (following paragraph). As an example, we examined the PIE-PUM2 results that were called from two sets of biological replicates, and the RNA variant calling results showed that 73% (2163 out of 2968) of PIE-PUM2 target sites were edited with a minimum of five reads (60x coverage or higher), and 97% (2891 out of 2968) were supported by at least three reads.

During variant calling, we performed several quality control steps to ensure the accuracy of the results: 1) We filtered out low-quality reads, mapped the reads to the reference genome, and removed PCR duplicates. 2) We filtered out the editing sites within homopolymers, in the vicinity of indels, or at the start/end of reads for potential misalignments or sequencing errors. 3) The following standard was used when calling C-to-U and A-to-I editing variants using JACUSA2: a) at least 5 reads covering each editing site; b) To exclude false negative, the editing rate was no less than 5% in PIE-RBP and at least twofold of that in APAD or APADcom control; and c) to exclude false positive, the log-likelihood z-score ≥ 4 was used (which can be lowered).

4. For the cellular experiments, the authors mentioned the transfection time was 24 h or 48 h. Did the authors estimate the optimal transfection time? How to evaluate the transfection efficiency?

The IRES-EGFP cassette in the PIE-Seq constructs was used to measure the transfection efficiency. At 48 hours after transfection, over 80% of the cells were EGFP positive; and at 24 hours after transfection, 30%~40% of cells were EGFP positive. We showed that expressing Apobec1 alone for 48 hours could introduce more stochastic C-to-U editing sites when compared to 24 hours (Fig. 2C-2D). Additionally, we observed that cells with weaker PIE-PUM2 expression displayed compromised editing on the positive control gene *CDKN1B* (Fig. S2B). Thus, we recommend using the 24-hour transient expression in combination with flow cytometry for PIE-Seq in cell lines.

5. In lines 130-156, the author compared the editing efficiency of joint the two editing proteins and only single ones, but the results of Figs. 2G-I showed that the jointly discovered target genes were not very prominent. After deducting the background information of ADAP, the result of PIE-PUM2 seemed to be less noisy. Does AD-PUM2 or AP-PUM2 also consider deducting the background information using only AD or AP? What will be the results?

Yes, AD and AP were already used as background control for AD-PUM2 and AP-PUM2, respectively in Fig. 2I.

6. Compared with PAR-CLIP-Seq method, very few targeted genes in the YTHDF family were identified by PIE-Seq. Reasonable explanations should be given for this result.

Indeed, it was intriguing that YTH proteins had fewer target genes in our assay. For YTHDF proteins, one potential explanation is that they promote stress granule formation which may affect deaminase activities (Fu and Zhuang, 2020). We have included this in the Discussion. Further works are required to understand this fully.

Minor Points:

1. In lines 97-99, the authors mentioned two controls for PIE-seq, but in the following analysis, only ADAP control was considered. Whether the empty control is necessary or not should be addressed.

The empty control is necessary: Editing sites in the empty control were excluded from downstream analysis. We have updated this detail in "Methods - Editing sites analysis".

2. In lines 102-103, the transfected cells were performed RNA-Seq for polyA mRNAs. The main recognition and binding sites of some splicing proteins (such as SRSF proteins) are on pre-mRNAs. Whether PIE-seq should consider the method of rRNA deduction for sequencing or not?

Thank you for this constructive suggestion! Indeed we used the rRNA deduction library preparation method for RBPs such as CELF1/2/4, FUBP1, KHDRBS1/2/3, NOVA1, TARDBP. We recommend rRNA deduction library preparation and have updated the Methods section.

3. When applying PIE-seq to single-cell sequencing, the 10-cell and single-cell samples showed higher C-to-U editing ratios than bulk samples, and more target genes were identified, which might be a rare case in single-cell sequencing. Possible validation assays for those identified target genes should be considered.

We observed higher editing rates in single cells in the MCP-MS2 assay than bulk samples (Fig. S4A), which is consistent with the higher target discovery in single-cell PIE-Seq (Fig. 4). We also observed higher variations in the editing rates of the positive control target *CDKN1B* among single cells than ten cells (Fig. 4B), suggesting higher variations of PIE-PUM2 expression among single cells. Additionally, the single/10-cell samples were sequenced at higher read depth (80~110 million reads per replicate), and analyzed with the same standards to define target sites and genes across all samples, which may have contributed to the increased number of identified target genes.

To mitigate this difference in read coverage between single-cell and bulk samples, we down-sampled the single-cell and 10-cell data and found that even with a 75% reduction (25% down-sampling) in read depth, we still identified over 67% and 63% of the target genes in 1-cell and 10-cell samples, respectively (new Fig. 4C and S4D). Target genes in 1-cell and 10-cell samples were significantly cross-validated with PUM2 PAR-CLIP targets (new Fig. S4E). These results suggest that specific single cells could show higher target discovery rates than bulk samples.

C

D

4. About the hyper-editing events which indicates 20 or more C-to-U editing sites per gene, is this standard defined by the authors or someone else? Is it necessary to normalize this standard by gene or mRNA length? This kind of information is missing in the manuscript.

The term "RNA hyper-editing" originally referred to extensive A-to-I conversions on RNA duplexes by ADARs, which differs from selective editing where only a limited number of A-to-I conversions occur in an mRNA (Nishikura et al., 1991; Scadden et al., 2001). Here we define the hyper-editing event as 20 or more C-to-U editing sites per gene based on overall editing site distribution (Fig. 2C). We have updated the manuscript accordingly.

5. The format of some pictures needs to be improved. For example, the dot diagram in Fig. 1B lacks the horizontal axis information, and label of y-axis should be percent (%); The legend needs a title to explain the specific meaning of the heat map in Fig. 2J-K and Fig. 7A.

We have updated x,y axis labels in Fig. 1B.

6. When comparing the results of the two treatment conditions, the significance test need to be mentioned, such as the result which is shown in Fig. S5D.

We have performed statistical tests and described the significance in Fig. 1C and Fig. 5E (previous Fig. S5D)

7. Generally, the presentation and clarity of some of the pictures should be improved to ensure that they are easy to be understood and aesthetically pleasing.

We have improved most figures for labels, fonts, and data plots.

Reviewer #3 (Remarks to the Author):

In this manuscript, Ruan et al. report usage of PIE-Seq using dual deaminases in RNA binding proteins (RBP)-RNA interactions. Briefly, authors introduced A-to-I and C-to-U deaminase domains onto the RBPs of interest. Authors show reproducibility of this method in human HEK293FT cells, developing mouse cortex using in utero electroporation and an option to study 25 RBPs efficiently. Furthermore, authors developed inducible PIE-seq. Overall, this is a strong study with elegant experimental design and results that supports conclusions. This will be an important addition to the field of unbiased screens of RBP-RNA interactions. I have only minor suggestion:

1) Authors should include statistics for Figure 1C.

Thank you for pointing this out. We have updated Figure 1C and included statistics.

** See Nature Portfolio's author and referees' website at www.nature.com/authors for information about policies, services and author benefits.

REVIEWERS' COMMENTS

Reviewer #1 (Remarks to the Author):

The authors have done a good job in revising the manuscript.

Reviewer #2 (Remarks to the Author):

The authors have addressed most of my concerns. I recommended its publication.

Response to Reviewers' Comments

The reviewers did not raise any further questions, and all reviewers recommended our study for publication.

REVIEWERS' COMMENTS

Reviewer #1 (Remarks to the Author): The authors have done a good job in revising the manuscript.

Reviewer #2 (Remarks to the Author): The authors have addressed most of my concerns. I recommended its publication.